# Low-rank and Orthogonal Space Enhance Learning Efficiency in Neural Networks without Gradient Backpropagation

## Abstract

The brain possesses highly efficient learning algorithms that have not been fully understood. The gradient backpropagation (BP) serves as a powerful tool for training artificial neural networks, but it diverges from the known anatomical and physiological constraints of the brain. Conversely, biologically plausible learning algorithms have efficiency limitations in training deep neural networks. To bridge this gap, we introduce a perturbation-based approach called low-rank cluster orthogonal (LOCO) weight modification. Theoretical analysis shows that LOCO provides an unbiased estimate of the BP gradient and achieves low variance in gradient estimation. Compared with some brain-inspired algorithms, LOCO keeps mathematical convergence guarantees and improves the efficiency. It can train the deepest spiking neural networks to date without gradient backpropagation, achieving state-of-the-art performance on several benchmark datasets and exhibiting the ability to overcome catastrophic forgetting. These findings suggest that biologically feasible learning methods can be substantially more efficient than previously believed. Furthermore, avoiding gradient backpropagation allows LOCO to achieve O(1) time complexity for weight updates. This opens a promising avenue for developing distributed computing systems that are more efficient than BP-based counterparts.

## 1 Introduction

Brain is a highly efficient parallel learning system. Its learning algorithms are capable to configure large network quickly to perform various tasks thereby enabling us to adapt to diverse environments. In addition, brain's algorithms can largely mitigate the problem of catastrophic forgetting, allowing us to learn continually. So far, the key components underlying brain's efficient learning is still not fully understood. Artificial neural networks provide a useful opportunity to study the capability and limitation of various learning algorithms. Currently, mainstream learning algorithms can be broadly categorized into two major classes: Backpropagation (BP)-based and nonBP-based Lillicrap et al. (2020); Bellec et al. (2020); Perich et al. (2018). BP and its variants achieve impressive results in a variety of machine intelligence tasks Sejnowski (2020); Pei et al. (2019); Ma et al. (2022). However, they rely on learning rules that are not biologically plausible Illing et al. (2019). Specifically, BP requires exactly symmetric forward and backward connections to allow the gradients to be propagated back, which also lead to weight transport problem Lillicrap et al. (2020) and update-locking problem Jaderberg et al. (2017). These requirements limit the parallelization in distributed computing systems, preventing the optimization from being performed in a truly parallel manner.

NonBP-based algorithms include Hebb-based, scalar feedback and vector feedback Lillicrap et al. (2020). Many of them are inspired by the brain and therefore are biologically plausible and suitable for parallel computing Flower & Jabri (1992); Crafton et al. (2019); Payeur et al. (2021). Nevertheless, current nonBP-based algorithms exhibit notable limitations in training neural networks. Specifically, these rules demonstrate satisfactory convergence only in shallow networks Adrien Journé (2023), often with difficulties in training deeper architectures Amato et al. (2019); Legenstein et al. (2005). Notably, node perturbation (NP) Widrow & Lehr (1990) is a biologically plausible algorithm with convergence guarantees. It adjusts the connection weights by multiplying information from the pre- and postsynaptic membranes with a global reward signal Fiete et al. (2007); Kornfeld

et al. (2020); Miconi (2017); Bouvier et al. (2018); Aronov et al. (2008); Ali et al. (2013). Although NP has the advantage of mathematically guaranteed convergence in learning, its efficiency decreases rapidly as the variance increases with the growing number of neurons in the network Malladi et al. (2024); Hiratani et al. (2022). Therefore, understanding the brain's high efficiency in learning remains an open question. Solving this could not only provide insights into brain's operation, but also be instrumental for designing AI systems that can learn more efficiently and, at the same time, avoid the limitation associated with BP, i.e., imperfect for large scale distributed computing.

Recently, empirical studies about neural networks in the brain have revealed two significant phenomena, including 1) that neural representation that needs to be separated are often organized orthogonal to each other Flesch et al. (2022); Xie et al. (2022), suggesting that synaptic modifications may be aimed at minimizing interference between different tasks Zeng et al. (2019); Duncker et al. (2020); and 2) brain dynamics are often manifested in low-dimensional manifolds Flesch et al. (2022); Perich et al. (2018); Goudar et al. (2023), suggesting that the learning happens in a low-rank parameter space. To the best of our knowledge, no prior work has incorporated low-rank and orthogonality constraints into non-BP algorithms. Additionally, there has been no theoretical analysis investigating the impact of orthogonality and low-rank constraints on learning efficiency.

In this work, we demonstrate that, within a low-rank setting, orthogonality constraints significantly reduce the variance of gradient estimates, increase the upper bound of the learning rate, and ultimately enhance convergence efficiency. We proposed the LOCO (Low-rank Cluster Orthogonal Weight Modification) algorithm, which is a NP-based learning rule that incorporates the low-rank and orthogonal learning principles. As a result, LOCO ensures 1) synaptic modifications occur within a low-dimensional subspace spanned by historical input information. Thus, rather than being modified independently, synaptic weights are adjusted in a highly structured and correlated manner; and 2) different subspaces are task-specific and mutually orthogonal. We analytically show that LOCO can converge more quickly than the conventional NP algorithm. In addition, we demonstrated empirically that LOCO is capable of training deeper spiking neural networks compared to other plasticity-based mechanisms, resulting in the state-of-the-art results on several benchmark datasets.

Our results demonstrate that the efficiency of biologically plausible learning algorithms can be substantially improved by implementing brain-inspired constraints, providing potential explanation for brain's efficient learning capability. In addition, as a learning rule that avoids gradient backpropagation, LOCO is better suited for efficient training on large-scale distributed computing systems compared to BP.

## 2 METHODS

### 2.1 LOW-RANK CLUSTER ORTHOGONAL WEIGHT MODIFICATION (LOCO)

LOCO is a NP-based learning algorithm. NP was proposed as a local optimization method that is biological feasible and can be implemented by distributed computing hardware Lillicrap et al. (2020). Similar to BP, NP calculates local derivatives of a global loss function to individual network connections, thereby executing the learning process. However, different from directly computing the derivatives in BP, in NP, small perturbations of activity are introduced to each node in the network to estimate the derivatives. Specifically, the derivatives of each weight are estimated by the product of 1) the equivalent changes in connection weights that would lead to the same activity perturbation, and 2) the changes in the loss function. i.e., the reward, caused by the perturbation. As a result, if a perturbation improves the performance, the weights are modified to steer the network activity towards the direction of the perturbation, and vice versa. The strength of NP resides on 1) it requires only a global reward signal and information of local activity, which is similar to the brain's learning mechanisms and suitable for parallel computing systems, 2) it is a unbiased estimation of the derivatives in BP, which guarantees that the learning process of NP can convergence anywhere BP can be implemented. However, due to the large variance in the gradient estimates caused by random activity perturbation, the NP algorithm exhibits poor convergence efficiency Malladi et al. (2024), leading to very slow learning speed even in shallow networks. Importantly, the convergence efficiency deteriorates as the network depth increases, making it less suitable for training more complex networks.

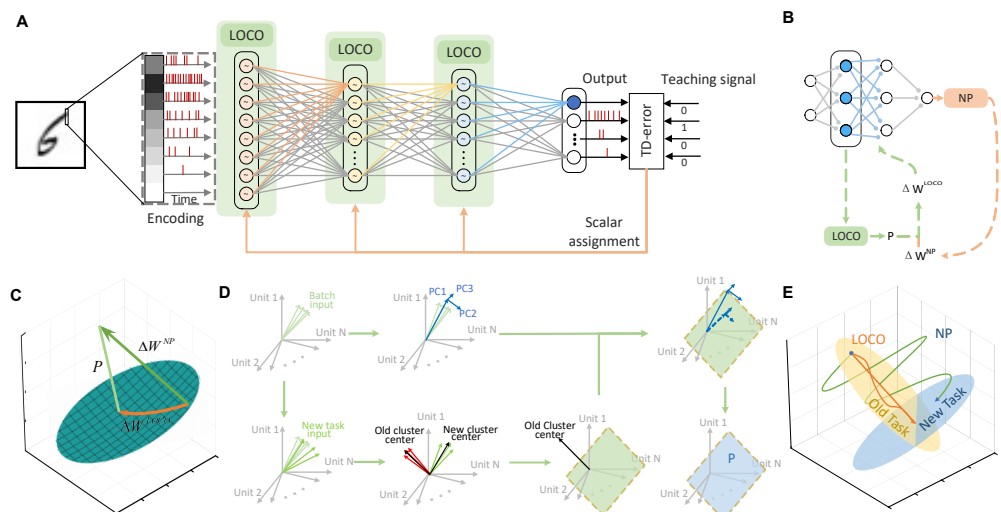

Figure 1: **Schematic diagram of LOCO.** **(A)** The multi-layer architecture of SNN, in which LOCO were introduced to modify the synapses at the hidden layers. The TD-error, which is the difference between the loss of two forward runs, was sent back to the hidden layers. The weights connecting to the same postsynaptic neurons are marked with the same color, representing the operation unit of weights in LOCO (Eq. 3). **(B)** Schematic diagram depicting the weight modification with NP and LOCO. NP suggests a direction for weights modification to reduce the loss. Then LOCO projects this direction to a lower dimensional subspace. Meanwhile, different task corresponds to separated subspace orthogonal to each other. **(C)** In the training process for new tasks, the original weight modification calculated by the standard NP, $\Delta \mathrm{W}^{NP}$, is projected to the subspace (dark green surface), in which good performance for old tasks can be achieved. As a result, the actually implemented weight modification is $\Delta \mathrm{W}^{LOCO}$. This process ensures that the weights configuration after learning the new task is still within the same subspace. **(D)** The process of calculating the projection matrix $P$ based on the historical input in LOCO. **(E)** Diagram showing the reduced variation during the parameter searching process between NP and LOCO. NP searches for parameters in a high-dimensional space (green line), with high variance. LOCO reduces the variance and facilitate learning by searching in a low-dimensional subspace (orange planes).

To solve the efficiency problem, here we propose the LOCO algorithm that can significantly reduce the variance in the learning process. In brief, LOCO introduces two extra constraints to the NP algorithm, i.e., the low-rank and orthogonality in the weights modification space, to greatly improve the efficiency of the learning process, while preserving the strength of local computation and provable convergence. Below we introduce LOCO and validate its effectiveness in Spiking Neural Networks (SNN) first and then show it can also be used to train Artificial Neural Networks (ANN).

In SNNs, we deployed a multilayer architecture constituted with Leaky Integrate and Fire (LIF) neurons, as illustrated in Fig. 1A, with the number of layers varying from three to ten. The learning process involves two forward propagation runs. The first run was the standard one. The second run introduced a random perturbation to the membrane potential of individual hidden layer neurons. The perturbations caused minor difference in the network's output values and the loss function compared to the first run. We name the difference in the loss function between the two runs as the temporal difference (TD) error, which was broadcasted to the hidden layer of the network to guide the learning (see Appendix for details). Specifically, based on the average firing rates of neurons in each layer and the TD error, the weight modification $\Delta W_l^{\mathrm{NP}}$ was obtained using the NP algorithm (Fig. 1B, details about mathematical basis of NP is listed in Appendix). Additionally, the average firing rates of neurons were also used to calculate LOCO constraints, which is a projection operator $P_l$ that restricts the weight modification to a subspace (Fig. 1C), resulting in $\Delta W_l^{LOCO}$ as the actual weight modification.

$$\Delta W_l^{LOCO\,T} = P_l \Delta W_l^{\text{NP}\,T} \tag{1}$$

LOCO imposes two constraints: 1) low-rank and 2) orthogonality to the space of weights modification ($S_w$), which were achieved by project the weights modification vector calculated by the NP algorithm $\Delta W_l^{\text{NP}}$ to a subspace using the orthogonal projector $P_l$ (Eq. 1).

We note that for the NP algorithm, $\Delta W_l^{\text{NP}}$ is always pointing to the same direction as the input vector X (Eq. 9). Thus, we can impose two constraints by reducing the space spanned by input vectors for category/task $i$ ($S_x^i$), and then applying $P_l$ to $S_x^i$ to obtain the direction of $\Delta W_l^{LOCO}$. Specifically, to impose the low rank constraint, the original $S_x^i$ was reduced to low dimensions, such as three dimensions of PC1-3, which were calculated using the principal component analysis (Fig. 1D upper row, see also Eq. 6). As a result, the variance of gradient estimates is reduced. We provide analytical prove that it enhances the convergence efficiency compared to the original NP method (see Appendix for details).

Orthogonal constraint reduces interference between different categories/tasks, improving the stability of the network's learning process and also contributing to increase convergence efficiency. To impose the orthogonal constraint when learning the category/task $i$, we first find the input vector $\mathbf{x}_k$ for k-th category/tasks by k-means clustering, and then calculate the space orthogonal to the space spanned by all $\mathbf{x}_k (k \neq i)$ except the $\mathbf{x}_i$ (Fig. 1D lower row, see also Eq. 5). Finally, projecting the low dimension space $S_x^i$ onto this orthogonal space yielded the weight modification space under the LOCO constraints (Eq. 6). The weight modification space constraint is represented as orthogonal projector $P_l$. Taken together, by greatly reducing the parameter space for searching compared to the NP method, the LOCO algorithm lowers the variation in the training process, and limits interference between tasks. As a result, these effects increase the speed of convergence (Fig. 1E).

## 2.2 Synaptic plasticity rule

LOCO extends the NP framework by incorporating orthogonal and low-rank constraints. These constraints ensure that the modifications to the weights occur within an orthogonal and low-dimensional space. The implementation of these constraints is achieved through projection operators, which project the weight modifications into an orthogonal, low-rank space.

$$\Delta W_l^{LOCO} = -\frac{\eta}{\sigma}(\tilde{\ell}(\tilde{s}^L, s^0) - \ell(s^L, s^0))\xi_l(P_l\mathbf{x}_{l-1})^T \tag{2}$$

The distinction between the weight update formula in LOCO and NP is the projection matrix prior to the input $\mathbf{x}_{l-1}$ in LOCO. $P_l$ denotes the orthogonal low-rank projection matrix for layer $l$. This formula is equivalent to $\Delta W_l^{LOCO\,T} = P_l \Delta W_l^{\text{NP}\,T}$. However, due to $\Delta W_l^{\text{NP}} \in \Re^{N \times N}$, $P_l \in \Re^{N \times N}$ where $N$ is the number of neurons in the hidden layer, the computational complexity of this calculation is $O(N^3)$. By using Eq. 2, and considering $\xi_l \in \Re^{N \times 1}$, $\mathbf{x}_{l-1} \in \Re^{N \times 1}$, the computational complexity is reduced to only $O(N^2)$.

$$\left[\Delta W_l^{LOCO}\right]_j = -\frac{\eta}{\sigma}(\tilde{\ell}(\tilde{s}^L, s^0) - \ell(s^L, s^0))[\xi_l]_j(P_l\mathbf{x}_{l-1})^T \tag{3}$$

The synaptics connected with $j$-th output neuron in $l$ layer $\left[\Delta W_l^{LOCO}\right]_j$ satisfied Eq. 3. The projection process can be viewed as operating on this set of synapses. It is the minimal unit for projection.

## 2.3 Calculation of projection matrix

The LOCO projection matrix constraint is derived in two steps. First, the Cluster Orthogonal Projection Matrix ($P_l^{CO}$) is computed. Then, the principal components of the input space are calculated and projected onto the Cluster Orthogonal space, eventually spanning a new space to yield the projection matrix $P_l^{LOCO}$ under the LOCO constraint.

The process to compute the $P_l^{CO}$ is referred to as the Cluster Orthogonal (CO) Projection Algorithm. The main idea of the CO Algorithm is to compute the projection matrix of the complement space of preserved directions $\mathbf{x}_k(k \neq i)$. This comes from the idea that if the direction of the weight

modification is orthogonal to all preserved directions, then this weight modification will almost not affect the input-output mapping of other categories, since the direction of the weight modification is the direction of the input for the current category (as Eq. 9). Hence, projecting the direction of the weight modification for the current category into the complement space means that the weight modification minimizes its impact on the weight modifications of other categories. The $P_l^{CO}$ is computed as follows

$$P_l^{CO} = I - A_l(A_l^T A_l)^{-1} A_l^T \tag{4}$$

where $P_l^{CO}$ is the orthogonal projection matrix. $A_l \in R^{n \times s}$ represents the preserved directions, which are the principal components of all categories except the current one. $n$ denotes the number of neurons in the hidden layer and also represents the dimensionality of the space in which the input resides. $s$ is the number of directions to be preserved, equivalent to the number of cluster centers minus one.

$$\begin{aligned}
U_{l-1} &= kmeans(X_{l-1}, c) \\
\mathbf{u}_{l-1}^j &= nearest(\mathbf{x}_{l-1}^p, U_{l-1}) \\
A_l &= \{U_{l-1}/\mathbf{u}_{l-1}^j\}
\end{aligned} \tag{5}$$

$X_{l-1} = [\mathbf{x}_{l-1}^0, \mathbf{x}_{l-1}^1, ..., \mathbf{x}_{l-1}^P] \in R^{n \times P}$ is a matrix composed of $P$ inputs. $\mathbf{x}_{l-1}^p$ denotes the principal component of the $p$-th batch in layer $l$, where each batch's data belongs to the same category. The k-means clustering algorithm is employed to cluster $P$ directions into $c$ cluster centers, represented by $U_{l-1} = [\mathbf{u}_{l-1}^0, \mathbf{u}_{l-1}^1, ..., \mathbf{u}_{l-1}^c]$. The $i$-th column of $A_l$ represents the principal component of the $i$-th category's input. However, this set of principal components excludes the principal component of the current category $j$, denoted as $A_l^i = \mathbf{u}_{l-1}^i, i \neq j$.

When the current input principal component is $\mathbf{x}_{l-1}^p$, the nearest cluster center $\mathbf{u}_{l-1}^j$ in $U_{l-1}$ is found in terms of angular distance,i.e., the $j$-th cluster center. By removing the $j$-th cluster center from all cluster centers, the final preserved directions $A_l$ are obtained. Through these operations, the Cluster Orthogonal Projection Matrix $P_l^{CO}$, which satisfies the orthogonality constraint, is derived.

Regarding the low-rank constraint, the first step involves identifying the top $k$ principal components of the current input batch. These top $k$ principal components are then projected into the space defined by the orthogonality constraint. Subsequently, a new space is spanned, and the projection matrix of this new space is obtained. The projection matrix of the newly formed space simultaneously satisfies both the orthogonality and low-rank constraints.

$$\begin{aligned}
Z_{l-1} &= PCA_k\left(X_{l-1}^{batch}\right) \\
Z_{l-1}^{CO} &= P_l^{CO} Z_{l-1} \\
Q_r &= Schmidt(Z_{l-1}^{CO}) \\
P_l &= P_l^{LOCO} = Q_r(Q_r^T Q_r)^{-1} Q_r^T
\end{aligned} \tag{6}$$

Where $X_{l-1}^{batch} \in R^{n \times batch}$ represents the data of one batch, with each data arranged as a column vector. The data within each batch belongs to the same category. $PCA_k(\cdot)$ denotes the operation of extracting the top $k$ principal components from the data. $Z_{l-1} \in R^{n \times k}$ represents these $k$ principal components. $Z_{l-1}^{CO} \in R^{n \times k}$ refers to the $k$ vectors obtained after projecting the $k$ principal components onto the Cluster Orthogonal constraint space.

The Gram–Schmidt process is then applied to $Z_{l-1}^{CO}$ for orthogonalization. After removing zero vectors, $Q_r \in^{n \times r}$ is obtained, where $r$ is the rank of $Z_{l-1}^{CO}$ and $r \leqslant k$. $P_l$ is the final LOCO constraint projection matrix derived from this process.

## 3 EXPERIMENTS

### 3.1 LOCO IMPROVES THE EFFICIENCY OF SNNs

To verify the effects the LOCO algorithm in training SNNs, we used three different learning tasks: (i) XOR task Payeur et al. (2021); (ii) recognition of handwritten digits using the Modified National

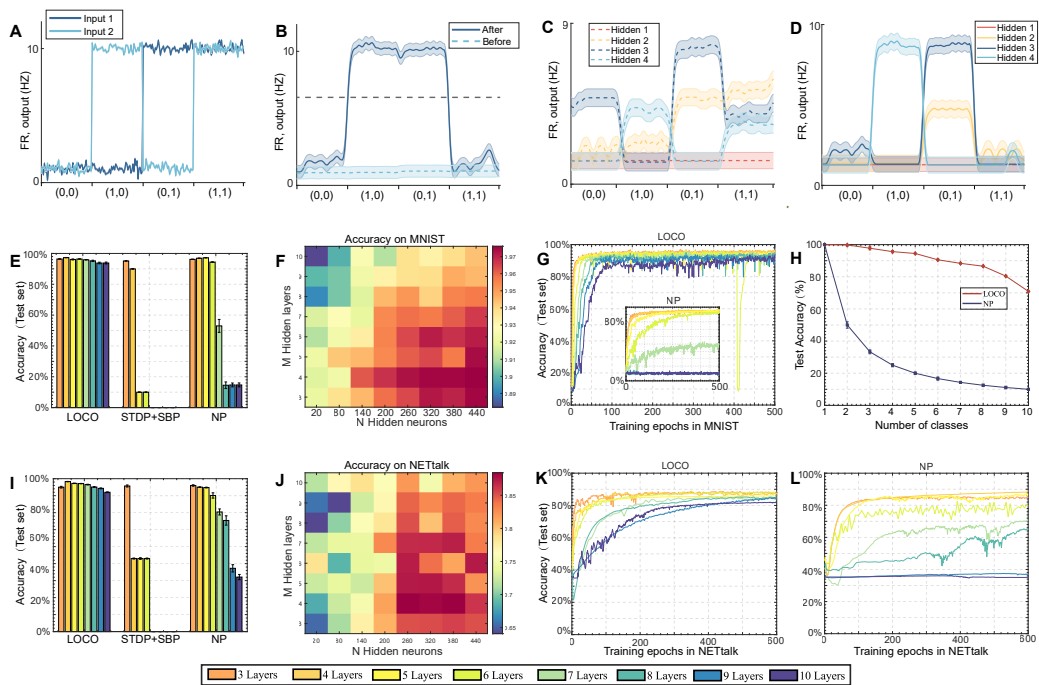

Figure 2: **LOCO can efficiently train SNN in various tasks. (A to D)** Performance of SNNs for the XOR task. **(A)** Input-layer signal for the four input conditions learned sequentially. FR, fire rate. The duration of each example was 8s.**(B)** Output signal of output layer neuron before and after learning the XOR task. **(C)** Fire rate of four hidden neurons before learning the XOR task. **(D)** Fire rate of four hidden neurons after learning the XOR task. **(E to H)** Performance of SNNs for the hand digit recognition task using the MNIST dataset. **(E)** LOCO can train a deeper spiking neural network than STDP+SBP and NP. Performance of network with different hidden layers are color-coded. **(F)** Accuracies of SNNs using pairs of superparameter number of hidden neurons (N) and the number of hidden layers (M). The accuracy is color-coded by the scale shown on the right. **(G)** Curve of accuracy with respect to the LOCO training epochs for neural networks ranging from 3 to 10 layers. The inset shows the curve of accuracy with respect to the NP training epochs for neural networks ranging from 3 to 10 layers. The SNN trained using LOCO converged faster and supported more layers compared to the one trained with NP. **(H)** Accuracy Curves for continual learning task on MNIST. The horizontal axis represents the current class being learned, with each class trained sequentially. The vertical axis indicates the average classification accuracy on all previously learned classes. The red curve shows the accuracy using LOCO, while the blue curve represents the accuracy using NP. **(I to L)** Performance of SNNs for the phonetic transcription task, presented in the same manner as that in (E) to (G).

Institute of Standards and Technology (MNIST) dataset LeCun (1998); and (iii) phonetic transcription using the NETtalk dataset Sejnowski & Rosenberg (1987).

XOR is a canonical example of a task that requires a nonlinear multi-layer network with appropriate credit assignment. For learning XOR task Payeur et al. (2021), we trained a 3-layer SNN, with 2 input neurons, 4 hidden neurons, and 1 output neurons (recorded as 1 when the firing rate is higher than 6 Hz). Our results showed that the XOR task can be learned using the LOCO algorithm (Fig. 2A, B), with the functional differentiation and specialization emerged for the hidden neuorns during learning(Fig. 2C, D).

For learning hand digit recognition on MNIST dataset, we use SNNs comprising 784 input neurons, 500 neurons per hidden layer, and 10 output neurons. The network depth in our experiments varies from 3 to 10 layers. We compare the highest accuracy achieved by LOCO, NP, and STDP+SBP algorithms across different network depths. The results demonstrates that LOCO could train networks with 10 layers, achieving an accuracy of 93.8% ± 0.15% (SD, $n$= 5 repeating experiments

with different random seeds) (Fig. 2E). By contrast, NP is only able to train up to 5 layers network with an accuracy of 90% ± 0.23% (SD, $n$= 5).

To examine the robustness of the LOCO algorithm to hyper-parameters, we vary the depth and number of hidden neurons. Fig. 2F shows that the algorithm can achieve good performance over a wide range of network parameters.

In addition, we find that the accuracy of SNNs trained by LOCO improve faster and reach a higher plateau compared to NP (Fig. 2G). For the best performance achieved by the NP algorithm (with 4 hidden layers, each one containing 500 neurons), the final test error rate was 5.26 ± 0.63% (SD, $n$= 5), and LOCO significantly reduced the error rate to 3.1 ± 0.21% (SD, $n$= 5) (Fig. 2E).

We also compare the performance of LOCO to the state-of-the-art algorithms using plasticity-based Diehl & Cook (2015), Wu et al. (2018), Zhang et al. (2018), Zhang et al. (2021a) and gradient-based Woźniak et al. (2020),Bellec et al. (2020), Jia et al. (2021) rules to train SNN. Compared to other plasticity-based algorithms, LOCO algorithm achieves the highest number of trainable layers in neural networks (with the criterion of reaching 0.9 highest accuracy). In addition, for SNN models with the same network parameter (network depth, the number of input, hidden and output neurons), the LOCO algorithms yields the best accuracy. The LOCO algorithms achieves good performance even when compared to some of the recently proposed gradient-based method (Table S1).

Beyond that, we also test the continual learning ability of the LOCO on MNIST (Fig. 2H). In a continual learning task, the network sequentially learns to recognize different sets of handwritten digits. It is conducted on a 3-layer feed-forward network with the same architecture configuration mentioned above. The results show that the NP algorithm suffers from catastrophic forgetting, since the performance rapidly declines when more new classes are learned. In contrast, LOCO exhibits stronger continual learning capability, showing much less interference among different tasks.

To examine the performance of LOCO in more complex tasks, we test its capability of phonetic transcription on NETtalk dataset, in which the network needs to classify 116 classes using 26 output neurons. In this task, we use an architecture comprising 189 input neurons, 500 hidden neurons per hidden layer, and 26 output neurons. We found that the LOCO algorithm exhibit advantages highly similar to that in the MNIST task. Specifically, compared to the NP method, it could train deeper networks (up to 10 layers) (Fig. 2I), shows the robustness to network parameters (Fig. 2J), lead to faster learning (Fig. 2K, L), and achieves better accuracy (Table S1). In addition, in the NETtalk task, the LOCO method also exhibits the best performance compared to the state-of-the-art plasticity-based methods and some recently proposed gradient-based learning methods (Table S1). More details about network models of SNNs, LIF neuron dynamic, data encoding method, network output and loss functions are listed in Appendix.

## 3.2 Unbiased gradient estimator and mitigating catastrophic forgetting

In the following experiments, we demonstrate that LOCO can also used for training ANNs. It aligns with BP in terms of gradient and has a stronger ability to overcome catastrophic forgetting. The overall procedure of applying the LOCO learning to ANNs is similar to that in SNNs, with some modifications made to accommodate characteristics of neurons in ANNs (see Methods for details). In addition to the three tasks tested for SNNs, a nonlinear fitting task is included to further demonstrate the consistency between the LOCO-estimated gradient and the BP-estimated gradient.

For learning the XOR task, we use ANNs with 2 input neurons, 4 or 50 hidden neurons, and 1 output neuron. We find that the convergence speed and final error rates of LOCO are similar to that achieved by BP, regardless of different network sizes (2-4-1 structure vs. 2-50-1 structure) (Fig. 3A, B). This indicates that LOCO does not experience a significant slowdown as the network size increases. Besides, the learning speed is significantly faster than training through NP.

We provide the proof that the weight change in LOCO, donated as $W$, is an unbiased estimator of the gradient in BP, donated as $B$ in the appendix of the paper. To empirically illustrate this property, we compare the weight changes in LOCO and the gradient in BP during the training of XOR. Figure 3C is the scatter plot of $W$ vs. $B$ at the beginning of training. Each point in the figure represents a weight change. The points predominantly reside in the first and third quadrants, indicating that $W$ and $B$ were mostly of the same sign, suggesting a general alignment in their overall direction. Throughout the training process, the angle between the weights vector trained by LOCO and the

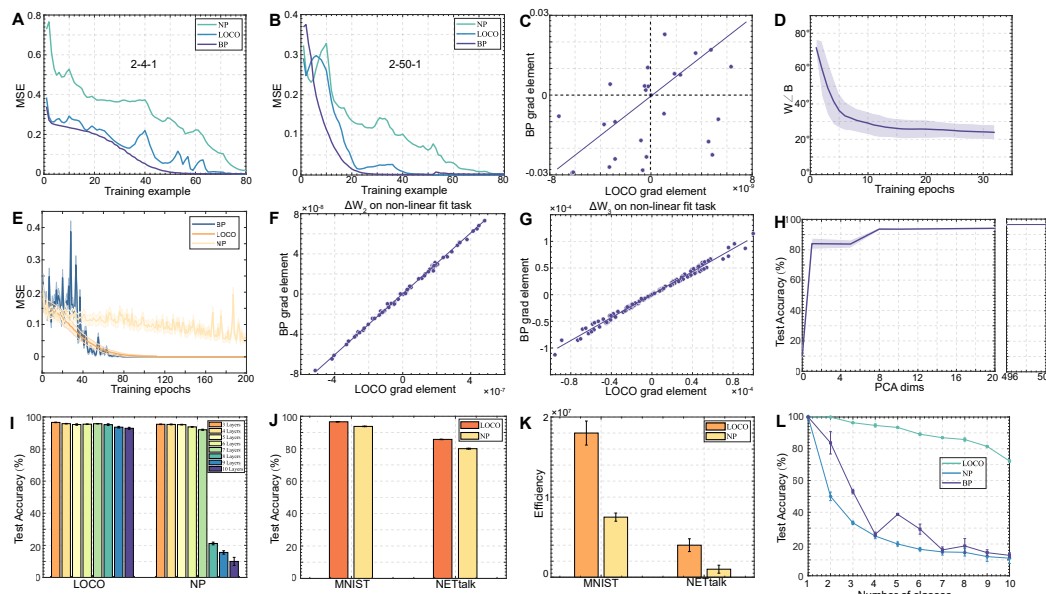

Figure 3: **LOCO can be also used to train ANN.** **(A to D)** LOCO training of an ANN to solve the XOR problem. **(A)** LOCO training approximates the performance of BP in the XOR task, in a small, three-layer (2–4–1) network. MSE, mean squared error. **(B)** Similar results as in (A) but with a wider, three-layer (2–50–1) network. **(C)** Scatter plot of the 25 corresponding elements of weight modifications $\Delta W^{LOCO}$ and weight gradients $\Delta W^{BP}$ following the training shown in (B). **(D)** The angle between weights trained using LOCO and BP changes throughout the training process. A smaller angle indicates better alignment in the direction of the weights trained by LOCO and BP. **(E to G)** Performance of ANN on a nonlinear fitting problem. **(E)** MSE loss of the BP, NP and LOCO algorithms with training epochs **(F-G)** Scatter plot between weight modification vectors in the second (F) and the third (G) layers trained by LOCO and BP, $\Delta W^{LOCO}$ and $\Delta W^{BP}$, exhibiting highly consistency. **(H to L)** Performance of ANN for the hand digit recognition task using the MNIST dataset. **(H)** The relationship between the highest achievable accuracy of a neural network and the low-rank dimensions constrained by PCA. **(I)** LOCO can train deeper spiking neural networks than NP. **(J)** The highest accuracy achieved by different algorithms on the MNIST and NETtalk datasets. **(K)** The convergence efficiency of the LOCO and NP algorithms when trained on the MNIST and NETtalk datasets. **(L)** Accuracy curves for continual learning task on MNIST. The horizontal axis represents the current class being learned, with each class trained sequentially. The vertical axis indicates the average classification accuracy on all previously learned classes.

final weights vector trained by BP decrease monotonically (Fig. 3D), demonstrating that the weights trained by LOCO and BP become more aligned.

In the nonlinear fitting task, the network is trained to fit $f(x) = \begin{bmatrix} \sin(x) \\ \sin(x) \end{bmatrix}, x \in [-\pi, \pi]$. The training set comprises 50 data points, uniformly distributed along the rang of $x$. A four-layer ANN with 1 input neuron, two 64-neuron hidden layers, and 2 output neurons is used. We find that in this task, the performance achieved by LOCO is essentially the same with BP, but LOCO method exhibits smoother learning curve than BP. In addition, the performance of LOCO is significantly improved compared to NP (Fig. 3E). Fig. 3F, G shows the scatter plot of $W$ vs. $B$ for the two hidden layers at the beginning of training, indicating high consistency in the estimation of gradients between LOCO and BP. We note that much larger range of $W$ and $B$ in Fig. 3F, G can better illustrate the consistency compared to that in Fig. 3C.

For learning hand digit recognition, we use ANNs with the same network structure as SNNs described above. In this task, we also test how the dimensionality constraint imposed to LOCO affect the learning. In theory, lower the dimensionality of the weight modification space may adversely affect the learning capability of the network. However, we discover that although the maximum ac-

curacy consistently improves as the dimensionality of the weight modification space increases, only a few key dimensions, identified through PCA, account for most of the learning capability. In other words, the majority of the improvement comes from just a small subset of important dimensions. (Fig. 3H). Specifically, for the classification task on the MNIST dataset, the network reaches good performance despite the dimensionality is decreased from 500 to 8. These result illustrates why the low-rank constraint in the LOCO algorithm did not significantly impair the learning capacity.

Similar to the results for SNNs, we can demonstrate that LOCO method is able to train much deeper networks of ANN on MNIST task (up to 10 layers, with the same network architecture as described in the SNN tasks) compared to the NP algorithm (limited to 5 layers)(Fig. 3I). In addition, with the MNIST and NETtalk tasks, we show that the LOCO method achieves better accuracy compared to NP (Fig. 3J).

We further compare the convergence efficiency of NP and LOCO with optimal hyper-parameters of ANN (hidden neurons and hidden layers). The result shows that the convergence efficiency for training with LOCO was substantially higher (58.3% reduction) than NP (Fig. 3K) for both MNIST (LOCO $[(0.75 \pm 0.051)\times10^7$; SD, $n$= 5], NP $[(1.8 \pm 0.15)\times10^7$; SD, $n$=5]) and NETtalk (LOCO $[(0.1 \pm 0.05)\times10^7$; SD, $n$= 5], NP $[(0.4 \pm 0.08)\times10^7$; SD, $n$=5]) tasks.

Finally, we test the continual learning capability of LOCO in ANN on MNIST(Fig. 3L). It is conducted on the 3-layer feed-forward network. LOCO results in much better performance compared with NP and BP algorithm, demonstrating its ability to overcome catastrophic forgetting also in ANNs.

## 3.3 LOCO LEARNING IN THE ORTHOGONAL SPACE

The orthogonal constraint introduced to the LOCO method aims to learn new categories with minimal disturbance to existing mappings. It is achieved by the orthogonal projector $P$, which restricts the weight modifications to the orthogonal space of the principal components of other categories (see Methods for the details regarding the construction of $P$). In the following analysis, we verify the orthogonal constraint empirically in the networks trained to preform the MNIST task.

First, we examines whether the weight modifications of the first category is orthogonal to others, by measuring the cosine similarity between the two. Fig. S1A shows that the LOCO algorithm indeed lead to smaller similarity compared to the NP algorithm, indicating that the weight modifications of other categories were mainly occur in the complement space of the first category.

The small cosine similarity of weight modifications between first and other categories is a reasonable result, indicating that weight modifications occur within the complement space of first category. These results suggest that the LOCO algorithm indeed constrains weight modifications to be orthogonal to the weight modification spaces of other categories during the learning process, fulfilling the orthogonal constraint. In contrast, the NP algorithm's weight modifications highly overlap with the spaces of other categories, leading to interference between categories.

Furthermore, we analyze the most sensitive weight when learning particular category and observe their changes across different categories. Fig. S1B shows that the weights most sensitive to a given category are uninvolved when learning other categories, suggesting the task specificity of weights and the orthogonal relationship between weight modifications across different categories.

The effects of the orthogonal constraint in alleviating the interference across different categories can be intuitively seen by looking at the learning curve of the system. Fig. S1C shows the accuracy in recognizing digit "1" along the training process. Compared with NP algorithm, LOCO indeed significantly reduces the adverse effect caused by learning other digits.

Finally, we analyze whether there is an orthogonal relationship at the level of neuronal activity, which has been reported in biological neural networks. Specifically, we compare the average cosine similarities, defined as $r$, to measure orthogonality at the activity level between different categories over the course of learning. Fig. S1D, E show that $r$ decreases with learning progress, suggesting that neural activities become orthogonal to each other between different categories. A direct visualization of activities of digit "0", "1", and "2" also illustrate the orthogonality in activities across different categories (Fig. S1F), similar to that has been observed in the biological brain Flesch et al. (2022).

### 3.4 LOCO LEARNING IN THE LOW RANK SPACE

The low-rank constraint introduced to the LOCO method aims to reduce the parameter search space during learning, thereby enhancing convergence efficiency. It is achieved by restricting the weight modifications to a low-dimensional subspace spanned by the first $k$ principal components of input vectors from the individual categories ($k \leq 20$ in the current study), which is much smaller compared to the original dimensionality of the original parameter search space, i.e., the number of neurons in the hidden layer (up to 500 in the current study). We provide the analytic proof that the low-rank constraint we introduced improves convergence speed compared to the NP method (see Appendix for details). In the following analysis, we empirically verify that the weight modifications are indeed constrained to the low-dimensional space in the networks trained to preform the MNIST task.

Fig. S2A shows that the rank of weight modification matrix during LOCO learning is much lower than that during the standard NP training, indicating that LOCO's weight modifications occur in a low-dimensional space. Analyzing the distribution of the ranks of weight modifications (Fig. S2B) during training also reveals that the LOCO ranks predominantly distribute in low values. Modifying weights in a low-dimensional space also reduces interference between different categories during training. Fig. S2C shows that modifying in low-rank space reduces the variation of network weights while maintaining accuracy, indicating that maintaining network stability has a significant impact on learning effectiveness. Concurrently, smaller magnitudes of weight adjustments imply that computational hardware can achieve a reduction in energy consumption. Fig. S2D shows that the weight modification of different categories reside in distinct subspaces. It can be proven that low-rank weight modification is also associated with low-rank network activity (see Methods for details), and the latter has been well documented in biological experiments Flesch et al. (2022).

## 4    DISCUSSION

There is a significant explanatory gap between the extraordinary learning capability of the brain and the low performance of brain-inspired, non-BP-based learning rules in training deep neural networks, which are often necessary to perform complex tasks.

Non-BP-based learning rules are effective only for networks with relatively fewer, e.g., no more than 5, layers. For instance, the Hebbian learning rule Pogodin & Latham (2020); Scellier & Bengio (2017); Millidge et al. (2022) is effective in shallow networks, faces challenge in maintaining stability and learning efficiency in deeper networks Amato et al. (2019); Legenstein et al. (2005). In a recent study, Zhang et al. proposed the Spiking Backpropagation (SBP) algorithm, extended the depth of optimizable layers in spiking neural networks from 2 to 4 in SNN Zhang et al. (2021a). Journé et al. proposed the SoftHebb, allowing effective training for up to five hidden layers Adrien Journé (2023). Although vector feed back learning rules have been applied to train networks with multiple linear hidden layers Frenkel et al. (2019), as multiple linear hidden layers are equivalent to a single linear layer, the networks trained in this way cannot achieve the same capability as deep non-linear networks Lillicrap et al. (2014). In this work, we proposed LOCO introducing the orthogonality and low-dimensionality in the process of training artificial neural networks. The relationship between LOCO and biological mechanisms is detailed in the Appendix.

LOCO suggests a promising path to develop efficient and reliable brain-inspired learning algorithms. By narrowing the explanatory gap between the brain's highly efficient learning and less efficient training of DNNs with non-BP methods, it shows that algorithms using simple scalar feedback for error signals can achieve efficient training of complex networks by introducing appropriate constraints on weight modifications. Removing the burden of complex back propagation procedure in calculating gradients makes local learning rules like LOCO better suited for efficient distributed computing. Furthermore, time complexity of feedback run during the LOCO training is $O(1)$, meaning that all weights can be modified simultaneously (see Appendix). Consequently, the feedback running time does not increase with the scale of the network, which is ideal for training large models.

## REPRODUCIBILITY STATEMENT

Our code, datasets, and checkpoints for the LOCO algorithm will be available at github after been accepted. Additionally, Appendix contains details about structures and hyperparameters for the models we trained.

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

# A APPENDIX

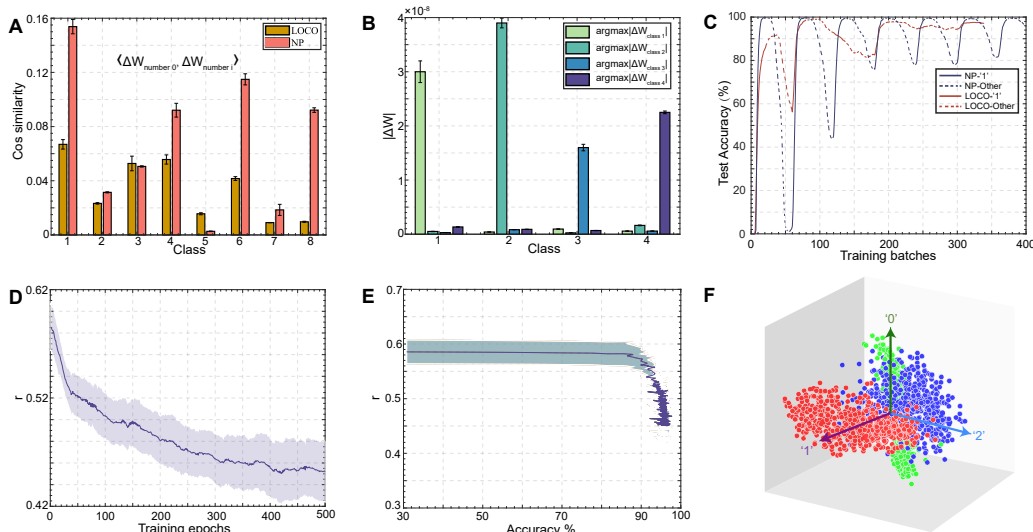

Figure S1: **LOCO restricts training in the orthogonal space.** **(A)** Weight modification implemented by LOCO is more orthogonal between different classes than that of NP. Each bar indicates the cosine similarity between weight modifications when learning the digit 0 and those made when learning the digits from 1 to 8. A lower cosine similarity suggests that the modifications in weights are more orthogonal to each other. **(B)** The most relevant weight modification in each class is selected. $argmax|\Delta W_{class\ i}|$ indicates the weights with the largest magnitude of change during the training of the $i$-th category. Bar graph indicates that each class has its own most relevant weight, suggesting that LOCO implements training in orthogonal space of weights. **(C)** Accuracy in recognizing digit 1 along the train process. LOCO exhibits smaller interference between training of different digits compared to NP. **(D)** The cosine similarities of activities between different categories $r$ (see Appendix) decreases during training, suggesting that neuronal activity tends to be more orthogonal along the course of training. **(E)** $r$ decreases as accuracy increases. This suggests that neuronal activity tends to be more orthogonal as accuracy increases. **(F)** Activity representations of three categories in the hidden layer during the training process. After dimensionality reduction via PCA, the activity representations of the three categories (color-coded) are positioned in a way tend to be orthogonal to each other. Each data point represents the average activity per batch.

| Method | Main rules | Accuracy | |
| --- | --- | --- | --- |
| | | **MNIST** | **NETtalk** |
| DiehlDiehl & Cook (2015) | STDP | 91.20±1.69% | —- |
| SOMHazan et al. (2018) | Hebb | 91.07±1.79% | —- |
| BBTZhang et al. (2018) | Balanced V+STDP | 93.67±0.40% | 84.26±0.20% |
| SNN-SBPZhang et al. (2021a) | STDP+SBP | 95.14±0.12% | 85.58±0.10% |
| BRPZhang et al. (2021b) | RP | 95.42±0.13% | 80.33±4.52% |
| NRRJia et al. (2021) | RP+BPTT | 94.19±0.11% | 77.73±0.46% |
| NP | Perturbation | 95.14±0.10% | 84.07±0.52% |
| LOCO | LOCO+Perturbation | **96.40±0.07%** | **86.01±0.35%** |
| LOCO (10layers) | LOCO+Perturbation | 93.80±0.12% | 82.00±1.95% |

Table S1: **The comparison of accuracy for different algorithms.** For algorithms without specific annotations regarding their parameter information, the quantity of parameters is identical. For the MNIST dataset, the network architecture is structured as 784-500-10. For the NETtalk dataset, the network configuration is 189-500-26.

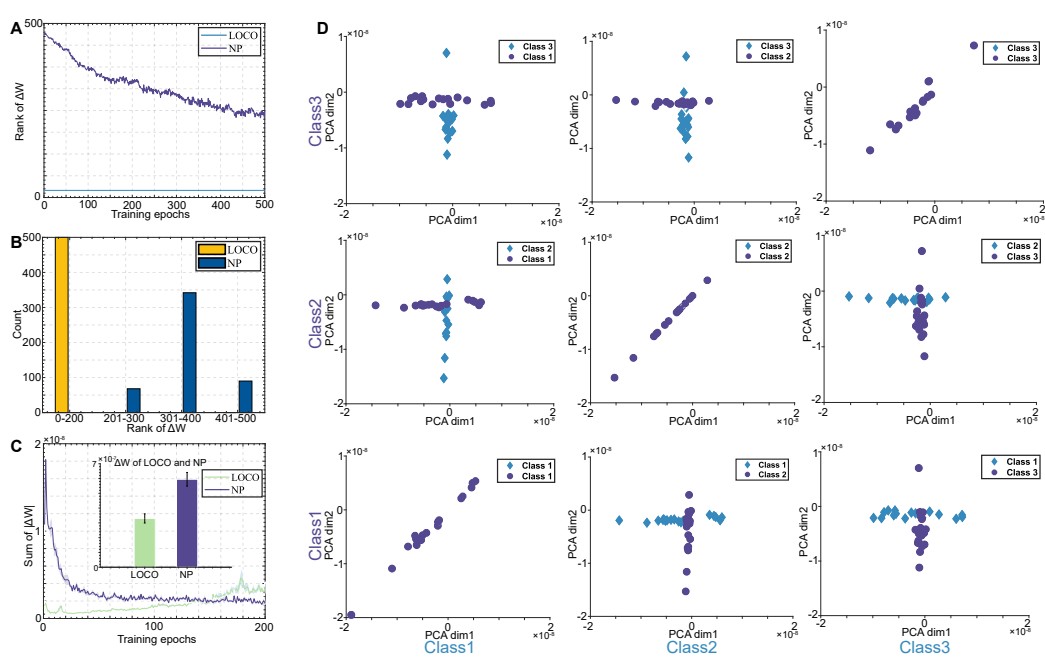

Figure S2: **LOCO restricts training in the low-rank space.** **(A)** Weight modification implemented by LOCO is in a subspace with lower rank than that of NP. **(B)** The distribution of rank of the weights modification space for LOCO and NP. **(C)** The magnitude of weight changes in LOCO is smaller than that in NP. This implies that LOCO is more stable and energy-efficient. **(D)** Weight modifications for recognizing a pair of digits (1-2, 2-3, and 1-3) are dimension reduced via PCA to be plotted in a 2D plane. The weight modifications for each class are clustered around a line in the plane, indicating that weight modifications are occurring within a low-dimensional space. In addition, weight modifications for different classes tend to be orthogonal to each other.

| Phases | Operation | Time Complexity |
|---|---|---|
| **Forward** | propagation | $O(bn^2)$ |
| | add direction $X$ | $O(bn)$ |
| | update $\mathbf{U}$ | $O(kpnc)$ |
| | nearest $\mathbf{u}_i$ | $O(bnc)$ |
| | delete $\mathbf{u}_i$ | $O(bc)$ |
| | calculate $P_l$ | $O(bc^2n + bc^3 + bnc^2)$ |
| **Backward** | weight grad | $O(bn^2 + bnc + bcn + bn^2)$ |
| | propagation | $O(bn^2)$ |

Table S2: **Time complexity of introducing projection matrix.** The time complexity of MLP is $O(bn^2)$ and the time complexity introduced by projection matrix does not exceed $O(bn^2)$. $b$ is batch size. $b = 32$ in all experiment settings. $n = 500$ is the number of neurons in the hidden layer. $p = 50$ is buffer size of direction $X$. $c = 10$ is the number of cluster centers and $k = 10$ is the number of iterations of k-means.

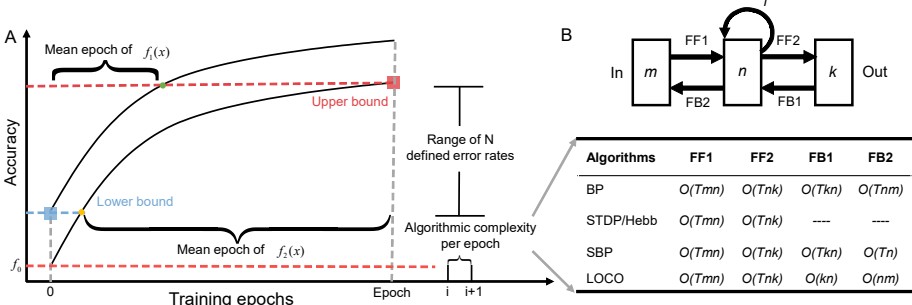

Figure S3: **The convergence efficiency during learning. (A)** Diagram depicting calculation of the mean epoch in N levels ($N = 5$) for curves of $f_1(x)$ and $f_2(x)$ to achieve some defined accuracy levels between an upper bound and a lower bound. The upper bound and lower bound represent the highest and lowest values of the accuracy curves at the beginning and the end of learning epochs, respectively, among the algorithms under comparison (see Methods for more details). The convergence efficiency was calculated by averaging the epochs at five accuracy levels (including upper and lower bounds). **(B)** Algorithmic complexity $O(\cdot)$ in each epoch during learning. It includes feedforward propagation (FF) and feedback propagation (FB). m, n, and k are numbers of neurons in network's input, hidden, and output layers, respectively. The compared algorithms include BP, STDP (or Hebb), self-BP (SBP) and LOCO.

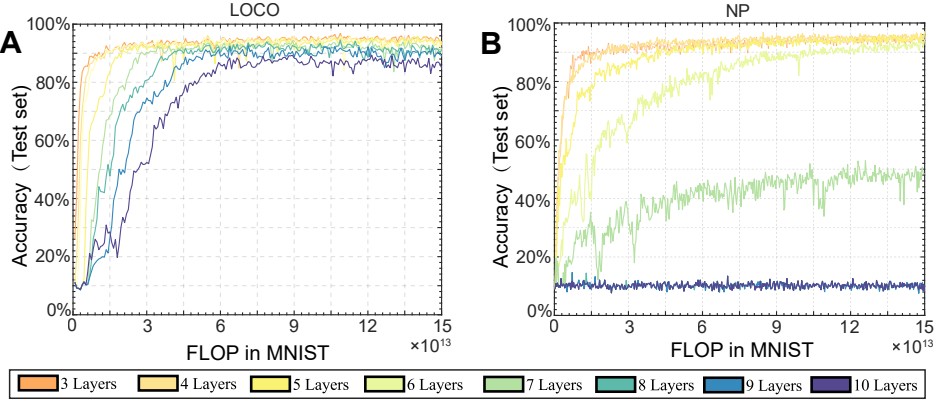

Figure S4: **FLOP efficiency during learning.** LOCO demonstrates faster convergence compared to NP across networks with 3 to 10 layers, highlighting its high efficiency when considering equivalent computational workloads. The FLOP calculation includes forward propagation, backpropagation, projection matrix computation, and the projection process for each layer.

## A.1 MATHEMATICAL BASIS FOR NP

Node perturbation is unbiased, but noisy. We first introduce and formulate NP in the context of deep spiking learning. Consider a deep feedforward network

$$s^l(t) = f_l\left(W^l s^{l-1}(t)\right), l = 1, 2, ..., L \tag{7}$$

where $s^0(t)$ and $s^L(t)$ are the input spike and the output spike respectively, and $f_l(\cdot)$ represents the propagation dynamics of the Leaky Integrate-and-Fire (LIF) neurons. which is an element-wise function. Adding a small Gaussian perturbation $\sigma\xi^l$ to each layer gives us a perturbed network.

$$\tilde{s}^l(t) = f_l\left(W^l \tilde{s}^{l-1}(t) + \sigma\xi^l\right), l = 1, 2, ..., L \tag{8}$$

where $\left\langle \xi^k \xi^{lT} \right\rangle = \delta_{kl} I_k$, with $I_k$ the identity matrix in the appropriate dimension, and $\sigma \ll 1$. Under the loss function $\ell(s^L, s^0)$ which is defined consistently with that in Eq. 27, the node perturbation update is

$$\Delta W_l^{\text{NP}} = -\frac{\eta}{\sigma}(\sigma\xi^l)(\tilde{\ell}(\tilde{s}^L, s^0) - \ell(s^L, s^0))\mathbf{x}_{l-1}^T \tag{9}$$

If the perturbation decreases the error (i.e., $\tilde{\ell} - \ell < 0$), the weights are shifted towards the direction of the perturbation ($\xi^l$), and vice versa. Importantly, in this update rule, the network only needs to know how much the loss changes when a perturbation is added to the network. This is in contrast to SGD and most of its biologically plausible variants, which require a supervised signal telling what the correct answer was. It is straightforward to show that, at the small perturbation limit $\sigma \to 0$

$$\tilde{\ell} = \ell + \sigma \sum_{l=1}^{L} \sum_{t=1}^{T} \frac{\partial \ell}{\partial h_{l,t}} \xi^{l,t} \tag{10}$$

where

$$h_{l,t} = W^l s^{l-1}(t) \tag{11}$$

Thus, denoting in the small limit the NP update becomes

$$
\begin{aligned}
\Delta W_l^{\text{NP}} &= \sum_{t=1}^{T} \Delta W_{l,t}^{\text{NP}} \\
&= \sum_{t=1}^{T} -\frac{\eta}{\sigma} \xi^{l,t} (\tilde{\ell}(\tilde{s}^L, s^0) - \ell(s^L, s^0)) s^{l-1}(t)^T \\
&= \sum_{t=1}^{T} -\frac{\eta}{\sigma} \xi^{l,t} \left( \sigma \sum_{k=1}^{L} \sum_{t'=1}^{T} \frac{\partial \ell}{\partial h_{l,t'}} \xi^{k,t'} \right) s^{l-1}(t)^T \\
&= -\eta \sum_{t=1}^{T} \xi^{l,t} \sum_{k=1}^{L} \sum_{t'=1}^{T} \frac{\partial \ell}{\partial h_{l,t'}} \xi^{k,t'} s^{l-1}(t)^T \\
&= -\eta \sum_{t=1}^{T} \sum_{k=1}^{L} \sum_{t'=1}^{T} \frac{\partial \ell}{\partial h_{l,t'}} \xi^{l,t} \xi^{k,t'} s^{l-1}(t)^T
\end{aligned} \tag{12}
$$

As mentioned above, taking expectation over $\xi$ gives us

$$\left\langle \Delta W_l^{\text{NP}} \right\rangle_\xi = -\eta \sum_{t=1}^{T} \frac{\partial \ell}{\partial h_{l,t}} s^{l-1}(t)^T \tag{13}$$

On the other hand, the SGD is given by

$$
\begin{aligned}
\Delta W_l^{SGD} &= -\eta \frac{\partial \ell}{\partial W_l} \\
&= -\eta \sum_{t=1}^{T} \frac{\partial \ell}{\partial h_{l,t}} \frac{\partial h_{l,t}}{\partial W_l} \\
&= -\eta \sum_{t=1}^{T} \frac{\partial \ell}{\partial h_{l,t}} s^{l-1}(t)
\end{aligned}
\tag{14}
$$

Therefore, NP is unbiased against SGD. Moreover, because SGD with i.i.d. samples is unbiased against the true gradient, NP is also unbiased against it. To simplify the model, the perturbation introduced in each layer during each propagation is identical.

## A.2  CONVERGENCE FOR NP AND LOCO

For other synaptic plasticity rules, there is often a lack of convergence guarantees at the network level. However, given that NP provide an unbiased estimation of gradients, convergence can be assured under conditions of problem convexity. The LOCO algorithm, derived from NP and augmented with additional constraints, does not alter the fundamental convergence properties of the algorithm. A rigorous proof of the convergence properties of NP is provided below.

The classical descent lemma uses a Taylor expansion to study how SGD reduces the loss at each optimization step. There is a Descent Lemma. Let $\ell(\theta)$ be convex and $\ell$-smooth. To simplify the expression, $\theta$ is subset of parameters with the same color like in Fig. 1A. $B$ is a batch of data. For any unbiased gradient estimate $\mathbf{g}(\theta, \mathrm{B})$

$$
\mathrm{E}[\ell(\theta_{t+1})|\theta_t] - \ell(\theta_t) \leq -\eta_{NP}\|\nabla\ell(\theta_t)\|^2 + \frac{1}{2}\eta_{NP}^2\ell \cdot \mathrm{E}[\|\mathbf{g}(\theta, \mathrm{B}_t)\|^2]
\tag{15}
$$

Unbiased gradient estimate means $\mathrm{E}[\mathbf{g}(\theta, \mathrm{B})] = \nabla\ell(\theta)$. The descent lemma also shows that to guarantee loss decrease, one needs to choose the learning rate as

$$
\eta_{NP} \leq \frac{2\|\nabla\ell(\theta_t)\|^2}{\ell \cdot \mathrm{E}[\|\mathbf{g}(\theta, \mathrm{B})\|^2]} = \frac{1}{r}\frac{2}{\ell} = \frac{1}{r}\eta_{SGD}
\tag{16}
$$

where $r$ is local r-effective rank. Sadhika Malladi et al. (2024) extensively discuss the convergence speed of NP in comparison to SGD (stochastic gradient descent), elucidating how the presence of low effective rank, denoted as $r$, ensures that NP do not suffer from undue slowness. This is attributed to the fact that the dimensionality of the search space required by the task, $r$, is significantly smaller than the number of model parameters $d$, i.e., $r \ll d$. Generally, the value of $r$ is task-dependent. This paper proposes that not only Hessian matrix has low effective ranks but also the gradient is low rank. We hypothesize that the upper bound for the ranks of both is $r$. The rank of the gradient is the dimensionality of the space spanned by the principal components of a certain class of input data. Empirical findings prove that the rank associated with the gradient are also markedly small (Fig. 3A, B, H).

Since NP provide an unbiased estimation of the gradient and represent the direction of steepest descent, the convergence properties of LOCO can be assessed by calculating the angle between the gradient estimated by NP and the weight modification direction in LOCO. For LOCO, the modification only involves the addition of a projection matrix $P_l$ in front of the gradient. This aspect makes it straightforward to demonstrate the convergence properties of the LOCO algorithm.

$$
\begin{aligned}
\cos\left\langle \Delta W_l^{\mathrm{np}}, \Delta W_l^{LOCO} \right\rangle &= \frac{tr\left(\Delta W_l^{\mathrm{np}} \cdot \Delta W_l^{LOCO}\right)}{\|\Delta W_l^{\mathrm{np}}\|_F \|\Delta W_l^{LOCO}\|_F} \\
&= \frac{tr\left(\Delta W_l^{\mathrm{np}} P_l \Delta W_l^{\mathrm{np}}\right)}{\|\Delta W_l^{\mathrm{np}}\|_F \|\Delta W_l^{LOCO}\|_F} \\
&> 0
\end{aligned}
\tag{17}
$$

Given that $P_l$ is a projection matrix, it is necessarily a positive definite matrix. Consequently, $\cos\left\langle \Delta W_l^{\mathrm{NP}}, \Delta W_l^{LOCO} \right\rangle > 0$, indicating that the angle between the gradient esti-

mated by NP and the weight modification direction in LOCO is within 90°. Moreover, as NP provides an unbiased estimation of the true gradient, the direction of weight modification in LOCO always forms an acute angle with the true gradient. This alignment continues to satisfy the conditions for convergence.

According to the Descent Lemma, it is still possible to arrive at the conclusion

$$\mathrm{E}[\ell(\theta_{t+1})|\theta_t] - \ell(\theta_t) \leq -\eta_{LOCO}\|\nabla\ell(\theta_t)\|^2 + \frac{1}{2}\eta_{LOCO}^2\ell \cdot \mathrm{E}[\|\mathbf{g}(\theta, \mathrm{B}_t)\|^2] \tag{18}$$

$P$ is a projection matrix. The descent lemma also shows that to guarantee loss decrease, one needs to choose the learning rate as

$$\eta_{LOCO} \leq \frac{2\|\nabla\ell(\theta_t)\|^2}{\ell \cdot \mathrm{E}[\|\mathbf{g}(\theta, \mathrm{B})\|^2]} = \frac{1}{r-o}\frac{2}{\ell} \tag{19}$$

see Eq. 31 for proof. $r$ denotes effective rank of loss. $o$ denotes the number of overlapped dimensionality between the complement of CO space and $r$-dimensional space. The value of $o$ ranges from 0 to $c-1$. $c$ denotes the dimensions kept by cluster orthogonal weight modification. Previous studies have discovered that the effective rank is significantly lower than the number of parameters in the model Malladi et al. (2024). This phenomenon also accounts for the higher practical convergence efficiency of NP compared to its theoretical optimization efficiency. The size of the effective rank is task-dependent. For instance, experiments on the MNIST task have revealed that the effective rank is approximately 20.

### A.3 PROOF OF UNBIASEDNESS OF LOCO

Let $\Delta W^{LOCO}(B_t)$ denote the LOCO weight update and $\Delta W^{NP}(B_t)$ denote the NP weight update for a given batch $B_t$. We have the relation:

$$\Delta W^{LOCO}(B_t) = P(B_t) \cdot \Delta W^{NP}(B_t) \tag{20}$$

where $P(B_t)$ is a projection matrix associated with batch $B_t$. Each projection matrix $P(B_t)$ is rank-deficient for any individual batch but, when accumulated across multiple batches, the matrices $P(B_1), P(B_2), ..., P(B_n)$ collectively span the full weight space, resulting in full rank coverage.

Proof

Consider an arbitrary training period with $n$ batches, $B_1, B_2, ..., B_n$, each associated with a projection matrix $P(B_t)$ applied to the NP weight update. The LOCO weight update across these $n$ batches is given by

$$\frac{1}{n}\sum_{t=1}^{n}\Delta W^{LOCO}(B_t) = \frac{1}{n}\sum_{t=1}^{n}P(B_t) \cdot \Delta W^{NP}(B_t) \tag{21}$$

We examine the expectation:

$$E\left[\frac{1}{n}\sum_{t=1}^{n}\Delta W^{LOCO}(B_t)\right] = E\left[\frac{1}{n}\sum_{t=1}^{n}P(B_t) \cdot \Delta W^{NP}(B_t)\right] \tag{22}$$

Since we assume that $P(B_t)$ independently projects in different directions across different batches, we can regard the cumulative effect of $P(B_t)$ as approximating the role of an identity matrix. Thus, we have $E[P(B_t)] \approx I$. where $I$ is the identity matrix in the weight space. Consequently, we can rewrite the expectation as

$$E\left[\Delta W^{LOCO}\right] \approx \frac{1}{n}\sum_{t=1}^{n} P(B_t) \cdot \Delta W^{NP}(B_t) = E\left[\Delta W^{NP}(B_t)\right] \tag{23}$$

Consequently $\Delta W^{LOCO}$ is approximately an unbiased estimator of gradient.

## A.4 Convergence speed for LOCO

The above analysis delves into the convergence properties of both NP and LOCO , providing upper bounds for the learning rates of each optimization algorithm. From this analysis, it is possible to establish a relationship between the upper bounds of the learning rates for these two optimization methods. This relationship elucidates how the constraints and modifications inherent in LOCO, relative to NP, influence the maximum permissible learning rates for ensuring convergence within these frameworks.

$$\eta_{LOCO} = \gamma \eta_{NP} \tag{24}$$

where $\gamma = \frac{r}{r-o} > 1$. In the MNIST experiments conducted for this paper for example, $r$ (effective rank) is approximately 10 to 30, and $r - o$ (the dimension of the projection space after LOCO constraints) is around 1 to 20. Consequently, the coefficient $\gamma$ is around 1.5 to 10. Therefore, the learning rate ensuring the learning rate of the LOCO algorithm is greater than that required for the NP algorithm. This implies that the convergence efficiency of the LOCO algorithm should also be higher than that of the NP algorithm. The specific proofs of their respective convergence rates are presented below.

Based on the formula Eq. 15,Plugging in $\mathrm{E}[\|\mathbf{g}(\theta, \mathrm{B}_t)\|^2] = \|\nabla\ell(\theta_t)\|^2 + \frac{1}{B}tr\left(\sum(\theta_t)\right)$ and selecting a learning rate $\eta < \frac{1}{\ell}$ yields

$$\mathrm{E}[\ell(\theta_{t+1})|\theta_t] \leqslant \ell(\theta_t) - \frac{\eta_{NP}}{2}\|\nabla\ell(\theta_t)\|^2 + \frac{\eta_{NP}^2\ell}{2B} \cdot tr\left(\sum(\theta_t)\right)$$

Since $\ell(\theta_t)$ is $\mu$-PL Malladi et al. (2024) satisfy $\frac{1}{2}\|\nabla\ell(\theta_t)\|^2 \leqslant u\left(\ell(\theta_t) - \ell^*\right)$, we get

$$\mathrm{E}[\ell(\theta_{t+1})|\theta_t] \leqslant \ell(\theta_t) - \eta_{NP}u\left(\ell(\theta_t) - \ell^*\right) + \frac{\eta_{NP}^2\ell}{2B} \cdot tr\left(\sum(\theta_t)\right)$$

Since $tr\left(\sum(\theta_t)\right) \leqslant \alpha\left(\ell(\theta_t) - \ell^*\right)$, we have

$$\mathrm{E}[\ell(\theta_{t+1})|\theta_t] \leqslant \ell(\theta_t) - \eta_{NP}u\left(\ell(\theta_t) - \ell^*\right) + \frac{\eta_{NP}^2\ell\alpha}{2B} \cdot \left(\ell(\theta_t) - \ell^*\right)$$

Altogether,

$$\mathrm{E}[\ell(\theta_{t+1})|\theta_t] - \ell^* \leqslant \left(1 - \eta_{NP}u + \frac{\eta_{NP}^2\ell\alpha}{2B}\right)\left(\mathrm{E}[\ell(\theta_t)] - \ell^*\right)$$

Choosing $\eta_{NP} = \min\left(\frac{1}{\ell}, \frac{uB}{\ell\alpha}\right)$, we obtain

$$\mathrm{E}[\ell(\theta_{t+1})|\theta_t] - \ell^* \leqslant \left(1 - \min\left(\frac{u}{2\ell}, \frac{u^2B}{2\ell\alpha}\right)\right)\left(\mathrm{E}[\ell(\theta_t)] - \ell^*\right)$$

Therefore we reach a solution with $\mathrm{E}[\ell(\theta_t)] - \ell^* \leqslant \varepsilon$ after

$$t \approx \max\left(\frac{2\ell}{u}, \frac{2\ell\alpha}{u^2B}\right)\log\left(\frac{\ell(\theta_0) - \ell^*}{\varepsilon}\right)$$
$$= O\left(\left(\frac{\ell}{u} + \frac{\ell\alpha}{u^2B}\right)\log\frac{\ell(\theta_0) - \ell^*}{\varepsilon}\right)$$

iterations.

By Eq. 15, LOCO with $\eta_{LOCO} = \gamma\eta_{NP}$ yields

$$
\begin{aligned}
&\mathrm{E}[\ell(\theta_{t+1})|\theta_t] - \ell(\theta_t) \leqslant \\
&\gamma\left[-\eta_{NP}\|\nabla\ell(\theta_t)\|^2 + \tfrac{1}{2}\gamma\eta_{NP}^2\ell \cdot \mathrm{E}[\|\mathbf{g}(\theta,\mathrm{B}_t)\|^2]\right]
\end{aligned}
$$

As in the proof for NP, choosing $\eta_{NP} < \frac{1}{\ell}$ yields

$$
\begin{aligned}
&\mathrm{E}[\ell(\theta_{t+1})|\theta_t] - \ell(\theta_t) \leqslant \\
&\gamma\left[-\tfrac{\eta_{NP}}{2}\|\nabla\ell(\theta_t)\|^2 + \tfrac{\gamma\eta_{NP}^2\ell}{2B} \cdot tr\left(\sum(\theta_t)\right)\right]
\end{aligned}
$$

Therefore under $\mu$-PL and the $tr\left(\sum(\theta_t)\right) \leqslant \alpha\left(\ell(\theta_t) - \ell^*\right)$ assumption we obtain

$$
\begin{aligned}
&\mathrm{E}[\ell(\theta_{t+1})|\theta_t] - \mathrm{E}[\ell(\theta_t)] \leqslant \\
&\gamma\left[\left(-\eta_{NP}u + \tfrac{\gamma\eta_{NP}^2\ell\alpha}{2B}\right)\left(\mathrm{E}[\ell(\theta_t)] - \ell^*\right)\right]
\end{aligned}
$$

$$
\begin{aligned}
&\Rightarrow \mathrm{E}[\ell(\theta_{t+1})|\theta_t] - \ell^* \leqslant \\
&\left(1 - \gamma\left(\eta_{NP}u + \tfrac{\gamma\eta_{NP}^2\ell\alpha}{2B}\right)\right)\left(\mathrm{E}[\ell(\theta_t)] - \ell^*\right)
\end{aligned}
$$

Choosing $\eta_{NP} = \min\left(\frac{1}{\ell}, \frac{uB}{\gamma\ell\alpha}\right)$ yields

$$
\mathrm{E}[\ell(\theta_{t+1})|\theta_t] - \ell^* \leqslant \left(1 - \gamma\min\left(\frac{u}{2\ell}, \frac{u^2B}{2\gamma\ell\alpha}\right)\right)\left(\mathrm{E}[\ell(\theta_t)] - \ell^*\right)
$$

Therefore we reach a solution with $\mathrm{E}[\ell(\theta_t)] - \ell^* \leqslant \varepsilon$ after

$$
\begin{aligned}
t &\approx \gamma^{-1}\max\left(\tfrac{2\ell}{u}, \tfrac{2\ell\alpha}{u^2B}\right)\log\left(\tfrac{\ell(\theta_0)-\ell^*}{\varepsilon}\right) \\
&= O\left(\gamma^{-1}\left(\tfrac{\ell}{u} + \tfrac{\ell\alpha}{u^2B}\right)\log\tfrac{\ell(\theta_0)-\ell^*}{\varepsilon}\right)
\end{aligned}
$$

iterations.

Consequently, based on the lower bounds of iteration counts for LOCO and NP, the relationship between the convergence times of the two algorithms can be deduced.

$$
t_{LOCO} \approx \gamma^{-1}t_{NP}
$$

Specifically, when $r$ is 30 and $c$ is 10, resulting in $\gamma$ is around 30/21, the LOCO algorithm is observed to be around 1.5 times faster than the NP algorithm. Empirically, this ratio was verified. Through Eq. 19, the size of $\gamma$ can be estimated. By analyzing the gradient information of the second layer in the 4-layer MNIST task, we obtained that $\gamma$ is [1.60 ± 0.21; SD, n= 5]. This is consistent with the results observed in the experiment (Fig. 2G).

## A.5 NETWORK MODELS OF SNNS

The architecture of the spiking neural network (SNN) is characterized by a multi-layered, fully connected structure. Following the encoding process, the spike information is fed into Leaky Integrate-and-Fire (LIF) neurons. Neurons in one layer are interconnected with the subsequent layer through fully connected synaptic. Upon the excitation of the LIF neurons, the emitted spike signals are weighted by the synaptic, forming postsynaptic membrane currents. These currents propagate to the neurons in the next layer, altering their membrane potentials. The propagation dynamics are defined by the following equation

$$I^l_{syn,i}(t) = \sum_j w_{ij} s^{l-1}_j(t) \tag{25}$$

The equation represents the propagation dynamics of spikes from layer $l-1$ to layer $l$ within a spiking neural network. Here, $I^l_{syn,i}(t)$ denotes the postsynaptic membrane current at time $t$ for the $i$-th neuron in layer $l$. $w_{ij}$ represents the synaptic weight between presynaptic neuron $i$ and postsynaptic neuron $j$. $s^{l-1}_j(t)$ represents the spike emitted at time $t$ by the $j$-th neuron in the preceding layer $l-1$.

### A.6 THE LIF PROPAGATION IN SNNS

The dynamic of LIF use clock driven LIFNode in spikingjelly Fang et al. (2023). The spikes in presynaptic neurons trigger postsynaptic potentials, which are dynamically integrated and generate spikes in the postsynaptic neuron when the firing threshold is reached. The membrane potential $V(t)$ is calculated as follows

$$\tau_m \frac{dV(t)}{dt} = -(V(t) - V_{reset}) + I_{syn}(t)$$

$$s(t) = \begin{cases} 1 & if(V(t) \geq V^{Tr}) \\ 0 & if(V(t) < V^{Tr}) \end{cases} \tag{26}$$

where $\tau_m$ is membrane time constant, $I_{syn}$ is the presynaptic current. The membrane potential $V(t)$ will be reset when crossing threshold $V^{Tr}$ and clamped to the resting potential $V_{rest}$.

### A.7 STATIC DATA ENCODING

We use a commonly used coding method: direct coding Wu et al. (2019); Rathi & Roy (2021) to encode the static images. Direct coding treats the first layer of the network as the coding layer. This approach significantly reduces the simulation length while maintaining accuracy. The direct encoding operation is divided into two steps for a normalized image $x \in [0,1]^{W \times H}$. First, the first layer of the network receives external stimuli and transforms them into a constant input current. Subsequently, this constant current is transformed into a spike sequence $\{0,1\}^{T \times (W \times H)}$ by LIF neurons, as described in Eq. 26.

### A.8 NETWORK OUTPUT AND LOSS FUNCTIONS

In the spiking neural network, the output of the network is decoded using a rate decoder. This involves calculating the average firing rate of the neurons in the output layer to determine the final output of the network. The loss function employed is the Mean Squared Error (MSE). In the LOCO algorithm, the feedback component is represented by a scalar Temporal Difference error(TD). This scalar quantifies the change in the evaluation metric across twice forward propagation.

The loss of neural network $l$ is calculated as follows

$$l = \frac{1}{2N} \sum_i \left( \frac{1}{T} \sum_{t=1}^{T} I_{spikes,i}(t) - y_i \right)^2$$

$$\tilde{l} = \frac{1}{2N} \sum_i \left( \frac{1}{T} \sum_{t=1}^{T} \tilde{I}_{spikes,i}(t) - y_i \right)^2 \tag{27}$$

Two forward propagations yield two distinct losses: one is the loss from precise propagation, denoted as $l$, and another is the loss following the introduction of perturbation, denoted as $\tilde{l}$. $\sum_{t=1}^{T} I_{spikes,i}(t)$ represents the output value of the $i$-th neuron in the output layer of the neural network, while $\sum_{t=1}^{T} \tilde{I}_{spikes,i}(t)$ represents the output value of the same neuron after the introduction of

the perturbation during the second propagation. $y_i$ symbolizes the target output of the $i$-th neuron in the output layer.

Temporal Difference (TD) error is utilized to guide the learning process in the LOCO algorithm. The computation formula for the TD error is as follows

$$TD = \tilde{l} - l \qquad (28)$$

The Temporal Difference (TD) error characterizes the change in the network's performance following the introduction of a perturbation, relative to its performance without the perturbation. A positive TD indicates that the perturbation has improved the network's performance, whereas a negative TD suggests a decrease in performance.

A.9  NETWORK MODELS OF ANNS

The architecture of an Artificial Neural Network (ANN) is characterized by a multi-layered, fully connected structure. The number of neurons in the input layer corresponds to the dimensionality of the input data. Information is propagated to the subsequent layer of neurons after being weighted by the synaptic. The neurons in the next layer aggregate this postsynaptic membrane information and use an activation function to determine the value transmitted to the subsequent layers of the neural network. In the algorithm proposed in this paper, the ANN undergoes two propagation passes. The propagation equation is defined by the following formula

$$\begin{aligned} \mathbf{x}_l &= f(W_l^T \mathbf{x}_{l-1}), l = 1, 2, ..., L \\ \tilde{\mathbf{x}}_l &= f(W_l^T \tilde{\mathbf{x}}_{l-1}) + \sigma\xi_l, l = 1, 2, ..., L \end{aligned} \qquad (29)$$

where $\mathbf{x}_{l-1} \in R^{n \times 1}$ represents the input data for layer $l$. $W_l \in R^{n \times n}$ denotes the weights of layer $l$, which correspond to synaptic strengths. The function $f(\cdot)$ signifies the activation function, with the ReLU (Rectified Linear Unit) function being utilized in this paper. $\mathbf{x}_l \in R^{n \times 1}$ indicates the output values of the neural network at layer $l$ during the first precise propagation. $\tilde{\mathbf{x}}_l \in R^{n \times 1}$ represents the output values of the neural network at layer $l$ during the second propagation, after the introduction of a perturbation.

It is important to note that the $\mathbf{x}_{l-1}$ mentioned here directly corresponds to $\mathbf{x}_{l-1}$ in Eq. 2. The weight update formula is the same as that in Eq. 2. Additionally, the proofs for convergence and the demonstration of convergence speed follow the same rationale as previously outlined.

A.10  DEFINITION OF CONVERGENCE EFFICIENCY DURING TRAINING

The convergence efficiency ($Effi_i$) of algorithm $i$ during training is determined by multiplying the mean number of epochs required to reach a specified accuracy level (Fig. S1A) with the algorithmic complexity per epoch, denoted as $O(n)_i$ (Fig. S1B). For the purpose of comparing two algorithms (where $i = 1, 2$), the convergence efficiency is evaluated using the following formula:

$$Effi_i = \frac{1}{N} \sum_{l=1}^{N} Argmin\left(f_i(x) = Acc_l\right) \times O(n)_i \qquad (30)$$

where $Argmin(\cdot)$ is the argument of the minimum, $f_i(x)$ is the accuracy curve with input epoch $x$, $O(n)_i$ is the algorithmic complexity with $n$ depicting the number of parameters, and $N$ is the number of predefined accuracy levels ($N = 5$). $Acc_l$ is selected out from a range of accuracy, with a upper bound of $Min(Max(f_1), Max(f_2))$, defined as the relatively lower maximal accuracy of $f_1(x)$ and $f_2(x)$, and also with an lower bound of $Max(Min(f_1), Min(f_2), f_0)$, defined as the relatively higher minimal accuracy among $f_1(x)$, $f_2(x)$, and an additionally predefined accuracy $f_0 = 0.8$ (the minimally acceptable accuracy).

A.11  MNIST AND NETTALK DATASETS

MNIST dataset LeCun (1998) comprises 60,000 training and 10,000 test samples, each with a size of 28×28 pixels. This dataset covers 10 classes of handwritten digits ranging from 0 to 9. NETtalk

dataset Sejnowski & Rosenberg (1987) consists of 5,033 training and 500 test samples. Each input sample represents an aligned English word through a 189-dimension vector, with individual letters encoded as one-hot vectors of 27 dimensions. Phonetic outputs are represented by multi-hot 26-dimension vectors, encompassing 21 sequential pronunciation features (e.g., "Labial", "Dental", "Alveolar", etc.) and five stress features (e.g., "¡", "¿", "0", "1", "2"). Excluding punctuation, the total number of phonetic representation classes with stresses is 116.

### A.12 ACCURACY DEFINITION

In our experiments, the accuracy of MNIST is defined as the number of correctly identifying samples dividing by the number of all samples. Different from it, the accuracy of NETtalk is defined as the cosine similarity distance of identified phonemes and real phonemes for the consideration of the multiphonemes in the same sample.

### A.13 DEFINITION OF $r$ FOR MEASURING ORTHOGONALITY AT THE ACTIVITY LEVEL

We definite the average cosine similarities between different categories as $r$ to measure orthogonality at the activity level. Small $r$ indicating that different tasks is orthogonal on the activity level. The definition is,

$$r = \frac{1}{10^2} \sum_{i,j}^{10} \frac{x_i \cdot x_j}{\|x_i\| \cdot \|x_j\|}$$

$x_i$ is the activity of hidden neurons when recognising $i$ category.

### A.14 RELATIONSHIP BETWEEN NEURAL ACTIVITY AND WEIGHT MODIFICATION

In the LOCO algorithm, the effect of low-dimensional and orthogonal activity dynamics in the brain can be explained as restricting synaptic modifications to occur within a low-dimensional and orthogonal space. This enhances the efficiency of perturbation-based optimization and improves the stability of the brain. We note that for the NP algorithm, $\Delta W_l^{\mathrm{NP}}$ is always pointing to the same direction as the input vector $x$ (Eq. 9). Therefore, if $x$ are low-dimensional and orthogonal to each other, the $\Delta W_l^{\mathrm{NP}}$ would hold the same property. Consequently, low-dimensionality and orthogonality of neural activity will lead to the same property of weight modification. Further more, with the analysis of Convergence speed for LOCO above, low-dimensionality will lead to higher convergence efficiency.

### A.15 TIME COMPLEXITY OF TRAINING PROCESS

Training process only requires weight adjustments as Eq. 2, if the computation of P is finished in the forward propagation. $\xi_l\left(P_l\tilde{\mathbf{x}}_{l-1}\right)^T$ has been calculated, the weight modification is just the multiply of TD error $\left(\tilde{\ell}(\tilde{s}^L, s^0) - \ell(s^L, s^0)\right)$ with $\xi_l^i(P_l\tilde{\mathbf{x}}_{l-1})^{j^T}$. Considering each weight as a computational unit, and based on Amdahl's Law, the maximum speedup rate S is $Ln^2$. The time complexity of the optimization process is only $O(1)$ (independent of the number of neural network parameters), meaning that all weights can be modified simultaneously. Consequently, the training time does not increase with the scale of the network. This method is suitable for distributed training of large models and can achieve a high degree of parallel efficiency.

$$S = 1/\left((1-a) + a/m\right)$$

where $a$ represent the proportion of the computation that can be parallelized, and $m$ the number of parallel processing nodes. During the training process, all weights can simultaneously compute the weight updates and execute the modifications, thus the proportion of parallel computation is 1, i.e., $a = 1$. Considering each weight as a computational unit, and taking a fully connected network as an example, the number of weights is $Ln^2$, where $L$ is the number of layers and $n$ is the number of neurons in each hidden layer. Therefore, $m = Ln^2$. Hence, the maximum speedup of the training process through parallelization is $Ln^2$.

## A.16 THE UPPER BOND OF LEARNING RATE IN LOCO AND NP

**Lemma 1** Let B be a random minibatch of size B. Then, the gradient norm of LOCO is

$$E\left[\left\|g^{LOCO}(\theta, \mathrm{B})\right\|^2\right] = (r - o) \cdot E\left[\left\|\nabla\ell(\theta, \mathrm{B})\right\|^2\right]$$

the gradient norm of NP is

$$E\left[\left\|g^{NP}(\theta, \mathrm{B})\right\|^2\right] = r \cdot E\left[\left\|\nabla\ell(\theta, \mathrm{B})\right\|^2\right]$$

$\theta$ is a unit set of parameters mentioned in Eq. 3. $r$ is r-effective rank. $o$ denotes the number of overlapped dimensionality between the complement of CO space and $r$-dimensional space.

**Proof of Lemma 1.** Similar with the proof in Malladi et al. (2024), We first note that in the $\sigma \to 0$ limit, we have

$$g(\theta, \mathrm{B}) = \frac{1}{B} \sum_{(x,y)\in\mathrm{B}} zz^T \nabla\ell(\theta, \{(x, y)\})$$

Taking expectation over the batch B and the $z$, we have $E[g(\theta, \mathrm{B})] = \nabla\ell(\theta)$, so $[g(\theta, \mathrm{B})]$ is an unbiased estimator of the gradient.

Computing the second moment, we get

$$E\left[g(\theta, \mathrm{B})\,g(\theta, \mathrm{B})^T\right] = \frac{1}{B^2} \sum_{(x_1,y_1),(x_2,y_2)\in\mathrm{B}} E\left[\left(zz^T\nabla\ell(\theta, \{(x_1, y_1)\})\right)\left(zz^T\nabla\ell(\theta, \{(x_2, y_2)\})\right)^T\right]$$

Let $u, v$ be two arbitrary vectors. We have that

$$E_z\left[zz^T uv^T zz^T\right] = uv^T$$

then

$$\begin{aligned} E_z\left[zz^T uv^T zz^T\right] &= E_z\left[z^{\otimes 4}\right](u, v) \\ &= \frac{3d}{d+2} Sym\left(I^{\otimes 2}\right)(u, v) \\ &= \frac{d}{d+2} \cdot u^T v \cdot I + \frac{2d}{d+2} \cdot uv^T \end{aligned}$$

Therefore

$$E\left[g(\theta, \mathrm{B})\,g(\theta, \mathrm{B})^T\right] = \frac{1}{B^2} \sum_{(x_1,y_1),(x_2,y_2)\in\mathrm{B}} \frac{2d}{d+2} \cdot E\left[\nabla\ell(\theta, \{(x_1, y_1)\})\,\nabla\ell(\theta, \{(x_2, y_2)\})^T\right]$$
$$+ \frac{d}{d+2} \cdot E\left[\nabla\ell(\theta, \{(x_1, y_1)\})^T \nabla\ell(\theta, \{(x_2, y_2)\})\right] I$$

Next, note that when $(x_1, y_1) \neq (x_2, y_2)$, we have

$$E\left[\nabla\ell(\theta, \{(x_1, y_1)\})\,\nabla\ell(\theta, \{(x_2, y_2)\})^T\right] = \nabla\ell(\theta)\,\nabla\ell(\theta)^T$$

and when $(x_1, y_1) = (x_2, y_2)$ we have

$$E\left[\nabla\ell(\theta, \{(x_1, y_1)\})\,\nabla\ell(\theta, \{(x_2, y_2)\})^T\right] = \nabla\ell(\theta)\,\nabla\ell(\theta)^T + \Sigma(\theta)$$

Therefore

$$\frac{1}{B^2} \sum_{(x_1,y_1),(x_2,y_2)\in \mathrm{B}} \frac{2d}{d+2} \cdot E\left[\nabla\ell\left(\theta, \{(x_1,y_1)\}\right)\nabla\ell(\theta, \{(x_2,y_2)\})^T\right] = \nabla\ell\left(\theta\right)\nabla\ell(\theta)^T + \frac{1}{B}\Sigma\left(\theta\right)$$

and plugging this yields

$$E\left[g\left(\theta,\mathrm{B}\right)g(\theta,\mathrm{B})^T\right] = \frac{2d}{d+2}\cdot\left(\nabla\ell\left(\theta\right)\nabla\ell(\theta)^T + \frac{1}{B}\Sigma\left(\theta\right)\right)$$
$$+\frac{d}{d+2}I\cdot\left(\|\nabla\ell\left(\theta\right)\|^2 + \frac{1}{B}tr\left(\Sigma\left(\theta\right)\right)\right)$$

we have

$$E\left[\|g\left(\theta,\mathrm{B}\right)\|^2\right] = d\cdot\left(\|\nabla\ell\left(\theta\right)\|^2 + \frac{1}{B}tr\left(\Sigma\left(\theta\right)\right)\right)$$
$$= d\cdot E\left[\|\nabla\ell\left(\theta,\mathrm{B}\right)\|^2\right]$$

For NP, $z = vec\left(\xi\mathbf{x}^T\right)$. Given that the directions of inputs from each category mainly resides within a subspace, which is a low-rank $r$-dimensional subspace. As a result, $d = N\cdot r$. Similarly, we have

$$d = \begin{cases} N\cdot r & , z^{NP} = vec\left(\xi\mathbf{x}^T\right) \\ N\cdot(r-o) & , z^{LOCO} = vec\left(\xi(P\mathbf{x})^T\right) \end{cases}$$

As mentioned in Eq. 3, when we just consider a unit set of parameters, we have

$$d = \begin{cases} r & z =^{NP} vec\left(\xi_j\mathbf{x}^T\right) \\ (r-o) & z = vec\left(\xi_j(P\mathbf{x})^T\right) \end{cases}$$

Finally,

$$E\left[\|g^{LOCO}\left(\theta,\mathrm{B}\right)\|^2\right] = (r-o)\cdot E\left[\|\nabla\ell\left(\theta,\mathrm{B}\right)\|^2\right]$$
$$E\left[\|g^{NP}\left(\theta,\mathrm{B}\right)\|^2\right] = r\cdot E\left[\|\nabla\ell\left(\theta,\mathrm{B}\right)\|^2\right] \tag{31}$$

$$\eta_{NP} \le \frac{2\|\nabla\ell(\theta_t)\|^2}{\ell\cdot\mathrm{E}[\|\mathbf{g^{NP}}(\theta,\mathrm{B})\|^2]} = \frac{1}{r}\frac{2}{\ell}$$
$$\eta_{LOCO} \le \frac{2\|\nabla\ell(\theta_t)\|^2}{\ell\cdot\mathrm{E}[\|\mathbf{g^{LOCO}}(\theta,\mathrm{B})\|^2]} = \frac{1}{r-o}\frac{2}{\ell} \tag{32}$$

**Definition 1** (Gradient Covariance). The SGD gradient estimate on a batch B has covariance $\sum\left(\theta\right) = B\left(E\left[g\left(\theta;B\right)g(\theta;B)^T\right] - g\left(\theta\right)g(\theta)^T\right)$.

### A.17 ORTHOGONALITY AND LOW-RANK STRUCTURES IN BIOLOGICAL NEURAL NETWORKS

In this work, inspired by Orthogonality and low-rank structures obtained through empirical and theoretical studies, we proposed LOCO introducing them in the process of training artificial neural networks. Our results confirmed that these brain-inspired constraints indeed significantly improved the learning capability and efficiency in deep neural networks, when the BP-calculated gradients are not available. Neuronal dynamics are organized at a low-dimensional manifolds have been actively studied in recent years. However, how such feature can facilitate learning in networks is not well understood. In the LOCO algorithm, the weights modifications are based on network activity patterns. As a result, the low-dimensional activity dynamics naturally leads to low-dimensional changes in connections (see Relationship between neural activity dynamics and weight modification

in mathematics in Appendix), which in turn can significantly reduce the exploration in the parameter space and facilitate learning.

Recent empirical studies have revealed that neural activities in the brain are restricted to low-dimensional manifolds and activity patterns exhibit orthogonal characteristics across different tasks, leading to the suggestion that these features may be the key to the brain's efficient learning.

Flesch et al. Flesch et al. (2022) explored how neural networks effectively encode multiple tasks through orthogonality, which is manifested by projecting the representations of different tasks onto mutually orthogonal low-dimensional manifolds. Theoretically, such orthogonal manifold designs allow different tasks to be independent in neural representation, enhancing learning efficiency in multitasking and reducing interference in learning different tasks. Indeed, functional Magnetic Resonance Imaging (fMRI) results show that human brains employs similar orthogonal representation patterns when processing different tasks, suggesting that the brain optimizes information processing using orthogonality to minimize interference between tasks. In addition, Libby and Buschman Libby & Buschman (2021) discovered that the brain reduces interference between sensation and memory of the same auditory stimuli by rotating sensory representations into orthogonal memory representations. Recording of neural activities in the auditory cortex of mice showed that neural populations represent sensory input and memory along two orthogonal dimensions. This transformation process, facilitated by "stable" neurons (maintaining selectivity) and "switching" neurons (reversing selectivity over time), effectively converts sensory input into memory. Model simulation also confirmed that the rotation dynamic and orthogonal representations can protect memory from sensory interference. These studies suggest that orthogonal representation play a significant role in reducing interference between tasks.

In the study by Flesch et al. Flesch et al. (2022), another discovery is the existence of low-dimensional, task-specific representations in human brains, particularly in the prefrontal areas. Neural encoding along task-irrelevant dimensions is compressed, yet this compression still retains the original space of the inputs. It suggests that the brain can handle multitasking and complex scenarios by encodes tasks through low-dimensional representations. In addition, the low-dimensional dynamics of neural population activities have been well documented in the areas of motor learning and motor execution. For example, Sadtler et al. (2014) investigated neural activity patterns in the primary motor cortex of rhesus monkeys during learning, revealed intrinsic constraints in the form of low-dimensional manifolds. Another study Perich et al. (2018) found that the brain uses low-dimensional neural population activities for rapid behavioral adaptation. In the premotor (PMd) and primary motor (M1) cortices , despite high-dimensional complexity in neuronal activities, the core functional connectivity remains stable. As a result, the activity of hundreds of motor neurons is represented in a low-dimensional manifold that reflects the covariance across the neuronal population. Pandarinath et al. (2018) discovered co-activation patterns among neurons. This means that even without observing all neurons, the brain's computational processes can be understood by analyzing a few key latent factors. This led to the proposal of the Latent Factor Analysis Dynamic Systems (LFADS) method. Applying this method to M1 and PMd also led to the discovery that the activities of a large number of neurons can be described by low-dimensional dynamics. In addition, Goudar et al.'s study Goudar et al. (2023) explored the acceleration of learning speed in neural networks when learning similar problems. They found that network activity patterns in the learning process are formed within the low-dimensional subspace of neural activity, and efficiency in learning similar tasks can be enhanced by restricting parameter exploration within this low-dimensional subspace.

