# OpenReview forum: "Efficient Learning in Neural Networks without Gradient Backpropagation"
_ICLR.cc/2025/Conference — ICLR 2025 Conference Withdrawn Submission_

### Official Review · Reviewer_8kH8 · 2024-11-01

**Soundness:** 2
**Presentation:** 3
**Contribution:** 2
**Rating:** 3
**Confidence:** 3

**Summary:**

This paper presents an improved training algorithm based on node perturbation. The main idea is to project the gradients to be altered into a low-dimensional space that is orthogonal to the direction of the original gradient changes, thereby accelerating the convergence speed of node perturbation.

**Strengths:**

The theoretical aspects of this paper are clearly articulated, which is one of its strengths.

**Weaknesses:**

However, I have two main concerns:

1. **Biological Plausibility of Node Perturbation Compared to Backpropagation**: In the work of Lillicrap et al., it is suggested that backpropagation can actually be implemented using a local Hebbian-like rule. The conclusion regarding its biological implausibility arises from three main points:
   - a. Backpropagation demands synaptic symmetry in the forward and backward paths.
   - b. Error signals are signed and potentially extreme-valued.
   - c. Feedback in the brain alters neural activity.

   It would be beneficial to clarify how node perturbation or the proposed LOCO method addresses these three issues to demonstrate its greater biological plausibility.

2. **Comparison with Other Learning Algorithms**: My second concern stems from the lack of comparisons with other learning algorithms, which challenges the novelty of this work. The related work section is insufficient; there exists a substantial class of non-backpropagation algorithms, such as:
   - Neural Sampling in Hierarchical Exponential-family Energy-based Models
   - Predictive Coding in the Visual Cortex: A Functional Interpretation of Some Extra-Classical Receptive-Field Effects

   Additionally, the concept of orthogonal projection has already been explored in related works, such as:
   - Hebbian Learning Based Orthogonal Projection for Continual Learning of Spiking Neural Networks.

   Furthermore, node perturbation has been extensively studied in the context of artificial neural networks, including models like the forward-forward model,
   - Scaling forward gradient with local losses
   - Gradients without backpropagation.
I hope the authors can include comparisons with these learning algorithms, as both experimental and theoretical evaluations are lacking in this paper. This may obscure the novelty of the proposed work.

**Questions:**

see weaknesses

---

> ### Author Response · Authors · 2024-11-24
>
> We thank the reviewer for the feedback, useful comments and appreciate the opportunity to provide an explanation regarding the reviewer’s weaknesses.
>
> > **W1:** Biological Plausibility of Node Perturbation Compared to Backpropagation: In the work of Lillicrap et al., it is suggested that backpropagation can actually be implemented using a local Hebbian-like rule.
>
> **A:** In [1, 2, 3, 4], Node Perturbation (NP) is well-established as biologically plausible. LOCO is built upon NP principles. However, our goal is not to propose a more biologically plausible algorithm but to enhance the efficiency of biologically plausible approaches. One significant advantage of biologically plausible algorithms like NP is that they do not suffer from the weight transport problem, which is a critical bottleneck in highly parallel systems. LOCO enabling simultaneous optimization of all weights, could achieve a time complexity of O(1) in brain-like or neuromorphic systems,which BP cannot achieve due to its inherent sequential dependencies.
>
> While Lillicrap et al.'s work demonstrates biological plausibility, their proposed algorithm does not focus on efficiency. Moreover, their thorough analysis of non-BP algorithms primarily focuses on approximating BP, without providing proofs of unbiasedness or convergence efficiency.
>
> > **W2:** Comparison with Other Learning Algorithms
>
> **A:** We compared multiple non-BP algorithms, including STDP, SOM, BBT, SNN-SBP, and NRR, as shown in Table S1. Regarding the [5,6] mentioned by reviewer, they are scalar feedback-based methods. In this regard, we included comparisons with BRP and NRR, which are also scalar feedback-based algorithms. The results are presented in Table S1.
>
> The  [7] mentioned by reviewer is synaptic plasticity-based methods. For this category, we compared STDP, SOM, BBT, and SNN-SBP  in Table S1. It is also worth noting that [7] employs an orthogonality-based approach. However, it designed for continual learning tasks and is theoretically unsuitable for retraining on previously learned tasks. In contrast, our method uses clustering-based orthogonality, which allows for flexible training of both new and previously learned tasks while maintaining stability. Detailed results can be found in Fig. S1C.
>
> The [8,9] mentioned by reviewer are forward-forward approach, which shares the same mathematical principles as WP [10]. Both are zero-order gradient optimization methods. The main drawback of zero-order gradient optimization is low efficiency. Previous studies have shown that WP is less efficient than NP [11]. Our method demonstrates higher efficiency than NP. In addition, [8] reduces variance by constraining the gradient through complex network architectures, our work focuses on reducing variance by constraining the input space x. These two approaches are complementary rather than contradictory and can coexist.
>
> We are conducting additional comparative experiments. Due to time constraints, we could not include these results and theoretical analyses in the current version, but they will be added in a future revision.
>
> [1] On the Stability and Scalability of Node Perturbation Learning
>
> [2] Model of birdsong learning based on gradient estimation by dynamic perturbation of neural conductances
>
> [3] An anatomical substrate of credit assignment in reinforcement learning.
>
> [4] Biologically plausible learning in recurrent neural networks reproduces neural dynamics observed during cognitive tasks.
>
> [5] Neural Sampling in Hierarchical Exponential-family Energy-based Models
>
> [6] Predictive coding in the visual cortex: a functional interpretation of some extra-classical receptive-field effects
>
> [7] Hebbian Learning Based Orthogonal Projection for Continual Learning of Spiking Neural Networks.
>
> [8] Scaling forward gradient with local losses
>
> [9] Gradients without backpropagation
>
> [10] Fine-Tuning Language Models with Just Forward Passes
>
> [11] Learning curves for stochastic gradient descent in linear feedforward networks

---

### Official Review · Reviewer_J9Cs · 2024-11-01

**Soundness:** 2
**Presentation:** 2
**Contribution:** 2
**Rating:** 5
**Confidence:** 4

**Summary:**

This paper addresses the well-known challenges of implementing backpropagation in neural systems and proposes an alternative learning rule, LOCO, to train neural networks. LOCO extends node perturbation (NP) by introducing two additional constraints—low-rank and orthogonality in the weight modification space—which significantly improve learning efficiency. Through empirical results on tasks like XOR, MNIST, and phonetic transcription, alongside theoretical analysis, the authors demonstrate that LOCO outperforms traditional NP and others.

**Strengths:**

•	The pursuit of biologically plausible alternatives to backpropagation is a significant and intriguing problem for computational neuroscientists, though it holds less interest for the ML community, where backpropagation is already effective and versatile.

•	The authors support their findings by combining empirical results on well-known benchmarks with theoretical analysis, which strengthens the validity of their approach (although the theoretical assumptions would benefit from clearer presentation within formal statements).

**Weaknesses:**

•	If biological plausibility is the main motivator for this work, the learning rule should be computable using known biological processes. However, the projection matrix P lacks clear justification in terms of how it could be implemented biologically.

•	A significant concern of this submission is the absence of a discussion on the limitations.

•	A closely related work by Duncker and Driscoll et al. (NeurIPS 2020), which also uses a projection matrix P to promote orthogonal subspace learning (minimizing interference in continual learning), was not cited.

•	The presentation is mostly clear; however, the title is overly general and does not clearly distinguish this work from the extensive literature on learning without backpropagation.

•	The theoretical analyses are not framed within formal Theorem statements, making it unclear what specific conditions or assumptions (e.g., convexity) apply.

•	Minor typos: mixing up of \citep vs \citet in the second-to-last paragraph of the introduction, as well as "wights" on line 102 and "perofmrance" on line 399.

**Questions:**

•	Can you demonstrate how the projection matrix P could be computed using known biological signals?

•	Does your learning rule perform effectively only when the task can be learned via low-dimensional orthogonal manifolds? How does the rank of the task influence performance, and does the task rank need to be known in advance?

•	Could you discuss how your method relate to Duncker and Driscoll et al. (NeurIPS 2020)?

---

> ### Author Response · Authors · 2024-11-24
>
> We thank the reviewer for the feedback, useful comments and appreciate the opportunity to provide an explanation regarding the reviewer’s weaknesses and questions.
>
> > **W1:** If biological plausibility is the main motivator for this work, the learning rule should be computable using known biological processes. However, the projection matrix P lacks clear justification in terms of how it could be implemented biologically.
>
> **A:** We appreciate the reviewer’s valuable feedback regarding the biological plausibility of the projection matrix. Orthogonality at the activity level has indeed been observed in biological systems. By orthogonalizing neural activities, we found that this approach significantly improves efficiency.
>
> The Hebbian rule proposed by [1] has been shown to achieve orthogonality, which supports the idea that some biological mechanisms may lead to similar effects. LOCO’s biological plausibility lies in incorporating biologically inspired low-rank orthogonal constraints into neural activities. We focus on the real performance. Projection matrix is suitable on theoretical analysis and experimental results demonstrating efficiency gains.
>
> While it is indeed feasible to implement orthogonality through Hebbian learning, we currently lack the capability to conduct a rigorous mathematical analysis of this Hebbian approach. To ensure that our findings are both robust and valuable, we chose to focus on methods that allow for comprehensive mathematical analysis alongside experimental validation. We believe this work lays a solid foundation for future research to further explore such mechanisms.
>
> > **W2: ** A significant concern of this submission is the absence of a discussion on the limitations.
>
> **A:** We appreciate your valuable feedback. We will include discussion on the limitations of our work in new version. This will also cover an analysis of the relationship between the projection matrix and biological plausibility.
>
> > **W3:** A closely related work by Duncker and Driscoll et al. (NeurIPS 2020), which also uses a projection matrix P to promote orthogonal subspace learning (minimizing interference in continual learning), was not cited.
>
> **A:** We will cite this paper.
>
> > **W4:** The presentation is mostly clear; however, the title is overly general and does not clearly distinguish this work from the extensive literature on learning without backpropagation.
>
> **A:** We have revised the titile to "Low-rank and Orthogonal Space Enhance Learning Efficiency in Neural Networks without Gradient Backpropagation". We highlight that, under the low-rank background, orthogonality constraints significantly reduce the variance of gradient estimates, increase the upper bound of the learning rate, and ultimately improve convergence efficiency.
>
> > **W5:** The theoretical analyses are not framed within formal Theorem statements, making it unclear what specific conditions or assumptions (e.g., convexity) apply.
>
> **A:** We just mentioned“As classical descent lemma, let ℓ(θ) be ℓ-smooth” We will add convexity in A.2.
>
> > **W6:** Minor typos: mixing up of \citep vs \citet in the second-to-last paragraph of the introduction, as well as "wights" on line 102 and "perofmrance" on line 399.
>
> **A:** Thank you for pointing out these issues. We have corrected them and appreciate it.
>
>
>
> [1] Hebbian Learning Based Orthogonal Projection for Continual Learning of Spiking Neural

---

> ### Author Response · Authors · 2024-11-24
>
> > **Q1:** Can you demonstrate how the projection matrix P could be computed using known biological signals?
>
> **A:** The same with W1.
>
> > **Q2:** Does your learning rule perform effectively only when the task can be learned via low-dimensional orthogonal manifolds? How does the rank of the task influence performance, and does the task rank need to be known in advance?
>
> **A:** We would like to clarify that. In our study, low-rank itself is not a necessary condition for improving efficiency. Our perspective is that low-rank enhances the efficiency of orthogonality constraints significantly, as demonstrated in Equation (24).
>
> We emphasize that "low-rank" is a relative concept. If the space constrained by orthogonality is sufficiently large, the actual rank of the task may not necessarily be low. For instance, in CIFAR-100, there are 100 classes. Orthogonality constraints can reduce 99 dimensions; however, r can still exceed 100. Despite this, the efficiency coefficient 𝛾 remains large, and LOCO can still achieve substantial efficiency improvements.
>
> In summary, while low-rank conditions can further enhance efficiency, LOCO does not rely solely on this property. Even in cases where the low-rank condition is not met, LOCO ensures 𝛾>1, thereby maintaining higher efficiency compared to methods based on non-orthogonal projections (NP).
>
>
> > **Q3:**  Could you discuss how your method relate to Duncker and Driscoll et al. (NeurIPS 2020)?
>
> **A:** Our work demonstrates that orthogonality and low-rank properties reduce the gradient variance in non-BP algorithms, thereby improving learning efficiency. This is fundamentally different from the focus of Duncker and Driscoll, who do not discuss non-BP scenarios. In the BP context studied in their work, gradients inherently have no variance, which means their findings are unrelated to the mechanisms we provide in our paper. For an intuitive understanding, please refer to Fig. 1E in our manuscript.
>
> Additionally, the implementation of orthogonality in our work differs from theirs. Duncker's method is designed for continual learning tasks and is theoretically unsuitable for retraining on previously learned tasks. In contrast, our method employs clustering-based orthogonality, which allows for flexible training of both new and old tasks, with the goal of ensuring stability. Detailed results can be found in Fig. S1C.
>
> We hope this addresses your question and clarifies the distinctions between our approach and theirs.

---

### Official Review · Reviewer_tNaz · 2024-11-03

**Soundness:** 2
**Presentation:** 2
**Contribution:** 2
**Rating:** 3
**Confidence:** 3

**Summary:**

This paper introduces a new learning algorithm, low-rank cluster orthogonal (LOCO) weight modification. LOCO improves on node perturbation by projecting the layer inputs into task-dependent orthogonal low dimensional space, effectively reducing the parameter space searched by node perturbation. LOCO’s performance is tested on MNIST and Netalk datasets, including a catastrophic forgetting task, for both spiking neural networks and artificial neural networks and found to be better than vanilla node perturbation.

**Strengths:**

- Interesting algorithm that has connections to neuroscientific data and the orthogonal activity subspaces found in e.g. motor control.
- Testing the algorithm’s performance across both SNNs and ANNs is a strength.
- Algorithm certainly improves over node perturbation.

**Weaknesses:**

- Abstract, statement about traditional brain-inspired algorithms. Convergence guarantees to what? Also this is an incorrect statement, node perturbation itself has guarantees and e.g. see https://arxiv.org/pdf/2110.10815 for FA.
- The proposed approach’s efficiency  is not tested (e.g. flops) and does not seem to be efficient as the projection matrix must be calculated at every update.
- Similar to efficiency aspects, the proposed approach trades off one aspect of biological plausibility for another (the cluster orthogonal projection algorithm on the inputs).
- The method seems to rely on a simple task structure. Namely that the inputs can be projected to low dimensions (without losing too much task relevant information) and orthogonalised using kmeans clustering (relying on data prototypes). Neither of these seems valid assumptions for complicated tasks, and the submission does not test on appropriately hard tasks such as those in bartunov 2018 (Assessing the Scalability of Biologically-Motivated Deep Learning Algorithms and Architectures for ANNs.)
- Figures are small and difficult to read at standard zoom. In particular Fig 1.
- Comparisons to BP are not always made

**Questions:**

Couple of typos:
- 102 and 156, typo weights
- 160 to obtained,
- 319 we varies
- 399 typo performance

---

> ### Author Response · Authors · 2024-11-24
>
> We thank the reviewer for the feedback, useful comments.  We are glad that the reviewer found that “Interesting algorithm connecting neuroscientific data and orthogonal activity subspace” “Testing is strength” and “improvements over node perturbation”.
> We appreciate the opportunity to provide an explanation regarding the reviewer’s weaknesses.
>
> > **W1:** Abstract, statement about traditional brain-inspired algorithms. Convergence guarantees to what? Also this is an incorrect statement, node perturbation itself has guarantees and e.g. see https://arxiv.org/pdf/2110.10815 for FA.
>
> **A:** Our original phrasing may have caused some misunderstanding. In the paper, we aimed to emphasize LOCO's ability to achieve low variance in gradient estimation. While we acknowledge that some brain-inspired algorithms also possess convergence properties, like NP, as explained in the Introduction, Method, and Appendix A.2. LOCO builds upon these foundations by significantly reducing variance, thereby enhancing convergence efficiency.
>
> To better convey this point, we have revised the statement as follows:
>
> " Theoretical analysis shows that LOCO provides an unbiased estimate of the BP gradient and achieves low variance in gradient estimation. Compared with some brain-inspired algorithms, LOCO keeps mathematical convergence guarantees and improves the efficiency. "
>
> We hope this clarification addresses your concern and better reflects the intended emphasis.
>
> > **W2:** The proposed approach’s efficiency is not tested (e.g. flops) and does not seem to be efficient as the projection matrix must be calculated at every update.
>
> **A:** In Section 2.2, we explained that the computational complexity of calculating the projection matrix and performing the projection is $O(n^2)$, which is equivalent to the complexity of forward propagation. Therefore, introducing the projection does not increase the overall computational complexity.
>
> Furthermore, the additional computation introduced by the projection does not exceed 3 times the baseline (we listed time complexity of each process in Table S2 in detail). Despite this, the benefits of the projection are significant. For instance, as shown in Fig. 3G, the efficiency of NP decreases significantly when optimizing neural networks with more than 6 layers and can even fail to optimize in some cases. In contrast, LOCO demonstrates high efficiency across experiments with neural networks ranging from 3 to 10 layers.
>
> To further account for the computational cost, we replotted the training curves using flop as the unit (Figure S4). The results confirm that LOCO remains more efficient than NP.
>
> > **W3:** Similar to efficiency aspects, the proposed approach trades off one aspect of biological plausibility for another (the cluster orthogonal projection algorithm on the inputs).
>
> **A:** We appreciate the reviewer’s valuable feedback regarding the biological plausibility of the projection matrix. Orthogonality at the activity level has indeed been observed in biological systems. By orthogonalizing neural activities, we found that this approach significantly improves efficiency.
>
> The Hebbian rule proposed by [1] has been shown to achieve orthogonality, which supports the idea that some biological mechanisms may lead to similar effects. LOCO’s biological plausibility lies in incorporating biologically inspired low-rank orthogonal constraints into neural activities. We focus on the real performance. Projection matrix is suitable on theoretical analysis and experimental results demonstrating efficiency gains.
>
> While it is feasible to implement orthogonality through Hebbian learning, we currently lack the capability to conduct a rigorous mathematical analysis of this Hebbian approach. To ensure that our findings are both robust and valuable, we chose to focus on methods that allow for comprehensive mathematical analysis alongside experimental validation. However, we believe this work lays a solid foundation for future research to further explore such mechanisms.
>
> [1] Hebbian Learning Based Orthogonal Projection for Continual Learning of Spiking Neural Networks.

---

> ### Author Response · Authors · 2024-11-24
>
> > **W4:** The method seems to rely on a simple task structure. Namely that the inputs can be projected to low dimensions (without losing too much task relevant information) and orthogonalised using kmeans clustering (relying on data prototypes). Neither of these seems valid assumptions for complicated tasks, and the submission does not test on appropriately hard tasks such as those in bartunov 2018 (Assessing the Scalability of Biologically-Motivated Deep Learning Algorithms and Architectures for ANNs.)
>
> **A:** We would like to clarify that. In our study, low-rank itself is not a necessary condition for improving efficiency. Our perspective is that low-rank enhances the efficiency of orthogonality constraints significantly, as demonstrated in Equation (24).
>
> We emphasize that "low-rank" is a relative concept. If the space constrained by orthogonality is sufficiently large, the actual rank of the task may not necessarily be low. For instance, in CIFAR-100, there are 100 classes. Orthogonality constraints can reduce 99 dimensions; however, r can still exceed 100. Despite this, the efficiency coefficient 𝛾 remains large, and LOCO can still achieve substantial efficiency improvements.
>
> In summary, while low-rank conditions can further enhance efficiency, LOCO does not rely solely on this property. Even in cases where the low-rank condition is not met, LOCO ensures 𝛾>1, thereby maintaining higher efficiency compared to methods based on non-orthogonal projections (NP).
>
> > **W5:** Figures are small and difficult to read at standard zoom. In particular Fig 1.
>
> **A:** We have made modifications of Fig1 as suggested.
>
> > **W6:** Comparisons to BP are not always made
>
> **A:** Our primary goal is to propose a brain-inspired method that solves the weight transport problem and enables parallel training. Since backpropagation (BP) does not satisfy these criteria, we only included a simple comparison with BP in the ANN experiments. Furthermore, as spiking neural networks (SNNs) are non-differentiable, BP cannot be used for optimization. Hence, BP is not always included in the experiments.
>
> Thank you again for your valuable feedback and thoughtful comments, which have helped us to further clarify and strengthen our work.

---

### Official Review · Reviewer_79Wt · 2024-11-04

**Soundness:** 1
**Presentation:** 3
**Contribution:** 2
**Rating:** 3
**Confidence:** 4

**Summary:**

In this manuscript, the authors propose a novel biologically plausible learning algorithm called LOCO. This algorithm is based on the node perturbation (NP) method but incorporates an update projected onto a low-rank activity subspace. The authors explore the theoretical properties of LOCO and demonstrate numerically that it outperforms vanilla NP and several previously proposed algorithms when applied to multi-layered spiking neural networks on the MNIST and NETtalk datasets.

**Strengths:**

The numerical experiments are thoroughly and clearly explained.
Moreover, the results convincingly demonstrate the benefits of the proposed algorithm over conventional NP for deep neural networks with more than five hidden layers.

**Weaknesses:**

There are several mathematical inaccuracies in the presented results that affect the soundness of the conclusions:

- The abstract states that "LOCO provides an unbiased estimate of the BP gradient," which appears to be incorrect. In the limit of infinite perturbations, the LOCO update becomes $\Delta W^{LOCO}_l = \Delta W^{NP}_l P_l^T = \Delta W^{BP}_l P_l^T$. Thus, unless the projection matrix $P_l$ is full-rank, LOCO introduces a bias relative to the BP gradient.

- In Line 894, $\ell$ is referred to as the Hessian matrix. However, according to Malladi et al. (2024), $\ell$ represents the smoothness parameter (the Lipschitz constant) of the loss function.

- Eq. 17: trace is missing from the Frobenius inner product.

- Eq. 18: While a matrix P satisfying this equation exists, P in this context is no longer a projection matrix.

- A.3: Inequalities are missing from the equations throughout the proof, making the arguments difficult to follow.

- L965-966: The origin of $tr [\sum(\theta_t)]$ is unclear, and the equation appears to be incorrect.

- L1221: Although $g$ follows a Gaussian distribution if the perturbation is Gaussian, the covariance is not necessarily the identity matrix.

- A 15: There seems to be confusion between subspaces in the activity space and the weight space. Even if the activity of pre-units resides in an r-dimensional space, the gradient g may not be constrained to an r-dimensional space.

- L1239-1241: Is it assumed that $\sigma^2 = || \mu_g ||^2$? If so, why should that be the case?

While the learning algorithm is motivated by biological plausibility, the method for obtaining the projection matrix $P_l$ lacks biological plausibility.

It is well known that orthogonalization prevent catastrophic forgetting (French, CogSci 1991; Farajtabar et al., AISTATS 2020), hence the presented result on continual learning is not surprising.

**Questions:**

How did you control the learning rate when estimating learning efficiency? Additionally, algorithmic complexity may be less relevant in inherently parallel systems, such as the brain, neuromorphic chips, and, to a lesser extent, GPUs. How does this affect your conclusion?

Wouldn’t it be more effective to consider low-rank perturbations of the postsynaptic units rather than projecting the presynaptic units into a low-rank space? What is the motivation or benefit of projecting $x_{l-1} $ to a low-rank space instead of adding perturbations in a low-rank space?

---

> ### Author Response · Authors · 2024-11-24
>
> We thank the reviewer for the feedback, useful comments and appreciate the opportunity to provide an explanation regarding the reviewer’s weaknesses and questions.
>
> > **W1:** The abstract states that "LOCO provides an unbiased estimate of the BP gradient," which appears to be incorrect. In the limit of infinite perturbations, the LOCO update becomes. Thus, unless the projection matrix is full-rank, LOCO introduces a bias relative to the BP gradient.
>
> **A:** Our unbiasedness applies to batches or even the entire dataset, as unbiasedness for a single data is not particularly critical in practice. We acknowledge the lack of a formal proof for batches in the original submission, and we have added supporting details in Appendix A.3. We appreciate your constructive feedback, which has helped us improve the manuscript.
>
> > **W2:** In Line 894, $\ell $ is referred to as the Hessian matrix. However, according to Malladi et al. (2024), $\ell $ represents the smoothness parameter (the Lipschitz constant) of the loss function.
>
> **A:** $\ell $ should be Lipschitz constant smoothness parameter of the loss function. We have modified it.
>
> > **W3:** Eq. 17: trace is missing from the Frobenius inner product.
>
> **A:** Thank you for pointing this out. The original formula only expressed the cosine similarity between vectors. We have revised the formula to represent the cosine similarity between matrices in Eq. 17.
>
> > **W4:** Eq. 18: While a matrix P satisfying this equation exists, P in this context is no longer a projection matrix.
>
> **A:** P is consistently a projection matrix throughout the paper. In the revised manuscript, we have improved the proof of γ, there is no misunderstanding any longer.
>
> > **W5:** A.3: Inequalities are missing from the equations throughout the proof, making the arguments difficult to follow.
>
> **A:** Sorry for that, we have added Inequalities.
>
> > **W6:** L965-966: The origin of $\Sigma \left( \theta  \right)$ is unclear, and the equation appears to be incorrect.
>
> **A:** We add the definition of trace in (Definition 1).
>
> > **W7:** L1221: Although g follows a Gaussian distribution if the perturbation is Gaussian, the covariance is not necessarily the identity matrix.
>
> **A:** We only state that g follows a Gaussian distribution without implying that the covariance matrix must be the identity matrix. Additionally, in the revised proof provided in the updated version, this issue no longer exists in  A.16.
>
> > **W8:** A 15: There seems to be confusion between subspaces in the activity space and the weight space. Even if the activity of pre-units resides in an r-dimensional space, the gradient g may not be constrained to an r-dimensional space.
>
> **A:** In Section A.13 of the original paper, Section A.14 of the new version paper, we clarified that the weight update ΔW is aligned with the activity x. Additionally, the backpropagation (BP) formula demonstrates that the gradient g is aligned with the input, residing in the same space.
>
> > **W9:** L1239-1241: Is it assumed that ${\sigma ^2} = {\left\| {{\mu _g}} \right\|^2}$? If so, why should that be the case?
>
> **A:** To simplify the expression and clearly illustrate the influence of r and o on γ, we made several assumptions. We acknowledge that this might have caused some confusion. To address this, we have revised the proof of γ, making the proof more transparent. Please refer to Appendix A16 for the updated proof.
>
> > **W10:** While the learning algorithm is motivated by biological plausibility, the method for obtaining the projection matrix P lacks biological plausibility.
>
> **A:** We appreciate the reviewer’s valuable feedback regarding the biological plausibility of the projection matrix. Orthogonality at the activity level has indeed been observed in biological systems. By orthogonalizing neural activities, we found that this approach significantly improves efficiency.
>
> The Hebbian rule proposed by [1] has been shown to achieve orthogonality, which supports the idea that some biological mechanisms may lead to similar effects. LOCO’s biological plausibility lies in incorporating biologically inspired low-rank orthogonal constraints into neural activities. We focus on the practical performance. Projection matrix is suitable on theoretical analysis and experimental results demonstrating efficiency gains.
>
> While it is indeed feasible to implement orthogonality through Hebbian learning, we currently lack the capability to conduct a rigorous mathematical analysis of this Hebbian approach. To ensure that our findings are both robust and valuable, we chose to focus on methods that allow for comprehensive mathematical analysis alongside experimental validation. We believe this work lays a solid foundation for future research to further explore such mechanisms.

---

> ### Author Response · Authors · 2024-11-24
>
> > **W11:** It is well known that orthogonalization prevent catastrophic forgetting (French, CogSci 1991; Farajtabar et al., AISTATS 2020), hence the presented result on continual learning is not surprising.
>
> **A:** We primarily address the inefficiency caused by the instability and high variance of non-BP algorithms, such as NP. Overcoming catastrophic forgetting is merely a byproduct of our approach. Furthermore, our method not only overcomes catastrophic forgetting but also enables the repeated training of new and old tasks (Fig. 1S1C), with the primary goal of improving the stability of the training process. A more intuitive understanding can be found in Fig. 1E.
>
> > **Q1:** How did you control the learning rate when estimating learning efficiency? Additionally, algorithmic complexity may be less relevant in inherently parallel systems, such as the brain, neuromorphic chips, and, to a lesser extent, GPUs. How does this affect your conclusion?
>
> **A:** Theoretically, LOCO allows for a higher upper bound on the learning rate compared to NP, which enhances convergence speed. Our analysis of time complexity is to ensure that no additional computational overhead is introduced, rather than improving efficiency through reducing time complexity. In our paper, we use time complexity in three instances:
>
> 1.	Section 2.2: This demonstrates that introducing the projection matrix does not increase the time complexity.
>
> 2.	Figure S3B: This illustrates that, compared to other SNN optimization algorithms, LOCO does not need to account for the time window dimension T during the optimization phase, which reduces time complexity. This reduction is the same across all computational systems.
>
> 3.	Appendix A.15: This explains that in fully parallel computing systems, such as the brain or neuromorphic chips, LOCO enables weight updates to be executed in a fully parallel manner, achieving O(1) time complexity. In contrast, BP cannot be implemented in the brain or neuromorphic chips, and it cannot achieve O(1) fully parallel updates.
>
> > **Q2:** Wouldn’t it be more effective to consider low-rank perturbations of the postsynaptic units rather than projecting the presynaptic units into a low-rank space? What is the motivation or benefit of projecting x to a low-rank space instead of adding perturbations in a low-rank space?
>
> **A:** Regarding the projection of the input space, please refer to the paper [2]. The orthogonality constraint is only meaningful when projecting in input space. Since the weights are aligned with the input space (as shown in A.14), constraining the dimensionality of the input space can effectively reduce the variance of the weights (as demonstrated in A.16).
>
> Additionally, the suggestion to add noise exclusively in a low-dimensional space is very compelling. However, implementing this approach could be challenging. Adding noise to which dimensions of x remains a problem, and we will consider it in our future work.
>
> Thank you again for your valuable feedback, which will help us refine our approach.
>
> [1] Hebbian Learning Based Orthogonal Projection for Continual Learning of Spiking Neural
> [2] Continual learning of context-dependent processing in neural networks

---

### Note · Authors · 2025-04-24

I have read and agree with the venue's withdrawal policy on behalf of myself and my co-authors.

---

### Meta-Review · Area_Chair_u2dd · 2024-12-24

**Metareview:**

This submission was reviewed by four reviewers, all of whom unanimously recommended rejection. While the proposed LOCO algorithm aims to offer a biologically plausible alternative to backpropagation, significant weaknesses were identified. Despite the authors’ extensive rebuttals, key concerns remained unresolved, particularly regarding the novelty and practical impact of the method.

**Additional Comments On Reviewer Discussion:**

Reviewers raised concerns about the validity of the theoretical claims, noting inaccuracies such as biased gradient estimates and incomplete or unclear proofs. The biological plausibility of the method was also questioned, as the projection matrix, central to the approach, lacked a convincing link to biological mechanisms. This undermined the paper’s key motivation. Additionally, the experimental benchmarks were considered too simplistic, failing to demonstrate scalability or generalizability to more complex tasks, and comparisons with relevant non-backpropagation algorithms were insufficient. Despite the authors’ extensive rebuttals, key concerns remained unresolved, particularly regarding the novelty and practical impact of the method.

---

### Decision · Program_Chairs · 2025-01-22

Reject